# FUDOKI: Discrete Flow-based Unified Understanding and Generation via Kinetic-Optimal Velocities

**Jin Wang**[*,1]    **Yao Lai**[*,1]    **Aoxue Li**[2]    **Shifeng Zhang**[2]    **Jiacheng Sun**[2]
**Ning Kang**[2]    **Chengyue Wu**[1]    **Zhenguo Li**[†,2]    **Ping Luo**[†,1]

[1]The University of Hong Kong    [2]Huawei Noah's Ark Lab

## Abstract

The rapid progress of large language models (LLMs) has catalyzed the emergence of multimodal large language models (MLLMs) that unify visual understanding and image generation within a single framework. However, most existing MLLMs rely on autoregressive (AR) architectures, which impose inherent limitations on future development, such as the raster-scan order in image generation and restricted reasoning abilities in causal context modeling. In this work, we challenge the dominance of AR-based approaches by introducing FUDOKI, a unified multimodal model purely based on discrete flow matching, as an alternative to conventional AR paradigms. By leveraging metric-induced probability paths with kinetic optimal velocities, our framework goes beyond the previous masking-based corruption process, enabling iterative refinement with self-correction capability and richer bidirectional context integration during generation. To mitigate the high cost of training from scratch, we initialize FUDOKI from pre-trained AR-based MLLMs and adaptively transition to the discrete flow matching paradigm. Experimental results show that FUDOKI achieves performance comparable to state-of-the-art AR-based MLLMs across both visual understanding and image generation tasks, highlighting its potential as a foundation for next-generation unified multimodal models. Furthermore, we show that applying test-time scaling techniques to FUDOKI yields significant performance gains, further underscoring its promise for future enhancement through reinforcement learning.

## 1 Introduction

Driven by the rapid progress of large language models (LLMs) [1–5], a new wave of large-scale multimodal models has emerged, delivering remarkable advances in the two fundamental pillars of artificial general intelligence (AGI): understanding [6–10] and generation [11–15]. Building on this momentum, a growing body of work [16–25] seeks to unify perception and synthesis within a single framework, introducing versatile multimodal large language models (MLLMs) that seamlessly integrate visual understanding with image generation.

In prior research, most MLLMs adopt the autoregressive (AR) architecture of standard LLMs, processing multimodal tokens sequentially from left to right for both understanding and generation tasks [26, 27]. While these MLLMs deliver strong performance across many multimodal tasks, their

---

[*] Equal Contribution

[†] Correspondence to: Zhenguo Li <li.zhenguo@huawei.com> and Ping Luo <pluo@cs.hku.hk>.

Project page: fudoki-hku.github.io.

39th Conference on Neural Information Processing Systems (NeurIPS 2025).

## Generation

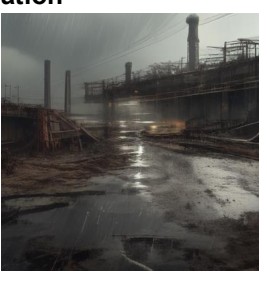

**A beautiful modern wooden house**, close to the lake, in the mountains at sunrise, **anime style**

**Hyper-realistic** photo of an abandoned **industrial** site during a **storm**

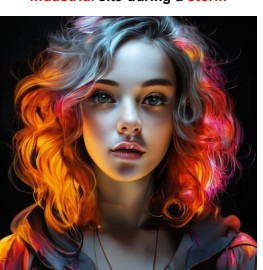

**Automobile** design drawings, **sketch**

Portrait of a beautiful **girl**, **neon**

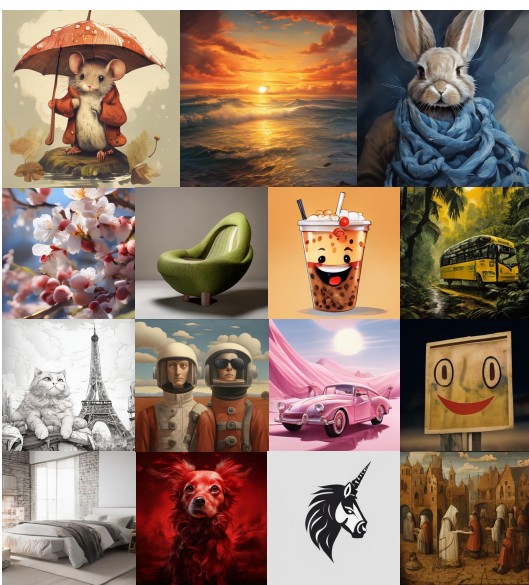

## Understanding

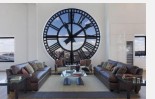

### 📦 Spatial Relation

**Question: What is the relation between the couch and the coffee table in the living room?**

A. The couch is on top of the coffee table
B. The coffee table is in front of the couch
C. The coffee table is behind the couch
D. The couch and the coffee table are not related

*Answer: The coffee table is in front of the couch.*

### 📈 Future Prediction

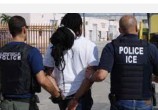

**Question: What is the anticipated outcome in this image?**

A. He will be released from the police station
B. He will escape from the police station
C. He will be arrested and taken to the police station
D. He will be visiting the police station voluntarily

*Answer: The image shows a person being escorted by police officers, with the person's hands behind their back. This indicates that the person has been arrested and is being taken to the police station. Therefore, the correct answer is C. He will be arrested and taken to the police station.*

### 🎞️ Scene Understanding

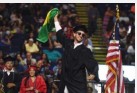

**Question: What is the main event that is happening in the image?**

A. An award show B. A concert performance
C. A graduation ceremony D. A sporting event

*Answer: The main event happening in the image is a graduation ceremony. This is evident from the individuals wearing graduation caps and gowns, and the presence of a large audience in a stadium-like setting, which is typical for graduation performance ceremonies.*

### 🔢 Instance Counting

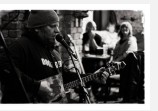

**Question: How many people are sitting in the background of the image?**
A. None B. One C. Three or more D. Two
*Answer: C. Three or more*

### ⚛️ Physics

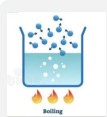

**Question: What is the transformation in this image?**
A. The water will freeze
B. The water will remain liquid
C. The water will evaporate
D. The water will condense

*Answer: The image shows a container of water being heated, with blue spheres representing molecules moving from the liquid to the gas phase. This process is known as boiling products, which transitions from a liquid to a gas state. Therefore, the correct answer is C. The water will evaporate.*

Figure 1: **Qualitative Results of Visual Generation and Understanding Capabilities of FUDOKI.** FUDOKI is designed based on the framework of discrete flow matching for both visual and textual modalities, capable of performing understanding and generation simultaneously under one unified paradigm.

inherent AR design's limitations have become increasingly apparent as shown in recent studies, such as weaker performance in complex reasoning [28–30], challenges in future planning [31], and difficulties with self-correction [32]. These shortcomings are particularly critical for emerging domains such as embodied AI and autonomous agents, where complex reasoning and deep contextual understanding are essential. This prompts a fundamental question for the future of AGI development: *what architectural paradigm could define the next generation of MLLMs?*

To this end, discrete-space generative flow and diffusion models have gained attention as a promising alternative for generative modeling. These models have seen success in the domain of text generation [33–38], protein design [39], image synthesis [37, 38], and code generation [37, 40]. Unlike sequential autoregressive models, these models usually begin with a fully corrupted sequence and iteratively denoise the entire sequence in parallel, which allows richer integration of information

from both directions to enhance prolonged reasoning. Moreover, these models enable flexible and controllable generation through their inherent iterative refinement process, while offering the potential for accelerated sampling via novel training designs [41–43]. Recent studies like LLaDA [44] and Dream [45] have also scaled discrete diffusion models to 7B parameters, further highlighting their growing potential to overcome the fundamental limitations of autoregressive approaches.

To advance the application of discrete generative flow modeling and challenge the dominance of the AR-based paradigm in MLLMs, we present FUDOKI, a unified multimodal model purely based on *discrete flow matching*. Different from previous diffusion-based unified multimodal models [46–48] focusing solely on the case of masking as a corruption process, we adopt the novel framework of discrete flow matching [37, 38], which substantially expanded the design space of discrete-space generative models by enabling metric-induced probability paths with kinetic optimal velocities. This design enables better performance than masked construction [38] and allows models to continuously self-correct their responses during the iterative refinement process. Moreover, to mitigate the high training cost of training large discrete flow matching models for multimodal tasks, we leverage the pre-trained AR-based MLLM [20] as the initialization and adaptively transfer it to the discrete flow matching paradigm [49].

The contributions of this paper can be summarized as follows: 1) We introduce FUDOKI[4], the first general-purpose unified multimodal model built entirely on discrete flow matching. Unlike traditional approaches that rely on masking-based corruption, FUDOKI leverages a metric-induced probability path with kinetically optimal velocities, expanding the design space of discrete multimodal modeling and offering advantages during inference; 2) Through extensive experiments, we show that FUDOKI achieves competitive performance on both visual understanding and text-to-image generation tasks, rivaling autoregressive-based MLLMs; 3) We apply test-time inference scaling techniques to FUDOKI inspired by [50], which yield substantial improvements across visual generation and understanding benchmarks. This suggests strong potential for future enhancement of FUDOKI via reinforcement learning [1, 51]. We believe that FUDOKI provides a compelling foundation for the development of next-generation unified multimodal models.

## 2 Preliminary: Discrete Flow Matching

In this section, we present key concepts and notations in discrete flow matching [37] to facilitate understanding in the following sections. Generally speaking, the objective of discrete flow matching is to approximate the target underlying data distribution $q(x)$ from the source known distribution $p(x)$, where $x = (x^1, x^2, ..., x^D)$ belongs to the discrete space $\mathcal{S} = \mathcal{T}^D$, where $D$ is the number of discrete variables and $\mathcal{T} = [K] = \{1, 2, \ldots, K\}$ represents a finite set of possible discrete values.

**Probability Paths**. Given a *source distribution* $p(x)$ and a *target distribution* $q(x)$ defined over a finite state space $\mathcal{S}$, discrete flow matching defines a family of time-indexed probability distributions $\{p_t(x)\}_{t \in [0,1]}$ to describe a smooth transformation from $p$ to $q$, referred to as *probability paths*. Each $p_t(x)$ is constructed as: $p_t(x) := \sum_{x_1 \in \mathcal{S}} p_t(x \mid x_1) q(x_1)$, where the conditional distribution is factorized across dimensions, namely $p_t(x \mid x_1) := \prod_{i=1}^{D} p_t(x^i \mid x_1^i)$. Here, each $p_t(x^i \mid x_1^i)$ defines a univariate interpolation between a base distribution $p(x^i)$ and a point mass $\delta_{x_1^i}(x^i)$, *i.e.*, $\delta_{x_1^i}(x^i) = 1$ if $x^i = x_1^i$ else 0. A common design for such interpolations is the *mixture path*, defined via a time-dependent scheduler $\kappa_t(x_1^i) \in [0, 1]$:

$$p_t(x^i \mid x_1^i) = (1 - \kappa_t(x_1^i))p(x^i) + \kappa_t(x_1^i)\delta_{x_1^i}(x^i), \tag{1}$$

where $\kappa_0(\cdot) = 0$ and $\kappa_1(\cdot) = 1$. This class of paths recovers the masked data construction when $p(x^i) = \delta_m(x^i)$ with $m$ denoting the *mask* token, which are widely used in previous studies [35, 36].

**Probability Velocities**. To simulate the generative process that evolves along the prescribed path $\{p_t(x)\}_{t \in [0,1]}$, we consider a continuous-time Markov chain (CTMC) $\{x_t\}_{t \in [0,1]}$ over the discrete space $\mathcal{S}$, such that: $x_t \sim p_t$. Specifically, we describe this CTMC via a *probability velocity* $u_t^i(\cdot, x_t)$

---

[4] 風土記 (*FUDOKI*) is a Japanese term referring to ancient records that comprehensively document and integrate the culture, geography, and traditions of different regions. We name our model *FUDOKI* to highlight its unified ability to both understand and generate multimodal information, such as interpreting and generating diverse images, mirroring how the original *FUDOKI* integrates and presents multifaceted knowledge.

(also known as the rate matrix), describing the rate of probability change of $x_t$ in its $i$-th token. Reminiscent of the velocity field in the continuous Flow Matching [42, 41], discrete flow matching features the following definition:

**Definition 1.** A probability velocity $u_t$ is said to generate the probability path $p_t$ if, for all $t \in [0, 1)$ and for any sample $x_t \sim p_t$, the updated sample $x_{t+h}^i \sim \delta_{x_t^i}(\cdot) + h u_t^i(\cdot, x_t)$ for each coordinate $i$ satisfies the condition that $x_{t+h} \sim p_{t+h} + o(h)^5$ as $h \to 0$.

Besides, the probability velocity $u_t$ should satisfy the following *rate condition*:

$$\sum_{x^i \in [K]} u_t^i(x^i, z) = 0, \quad \text{and} \quad u_t^i(x^i, z) \geq 0 \quad \forall i \in [D], \; x^i \neq z^i, \tag{2}$$

such that the updated $x_{t+h}^i$ can be sampled from a valid probability distribution. Further, previous studies [37, 39] also demonstrate the *Continuity Equation* (also known as the Kolmogorov forward equation) in discrete flow matching, which describes the state probability rate $\dot{p}_t(x)$, $x \in \mathcal{S}$ by:

$$\dot{p}_t(x) + \text{div}_x(p_t u_t) = 0. \tag{3}$$

where $\text{div}_x(p_t u_t) = \sum_{z \in \mathcal{S}} \sum_{i=1}^{D} \delta_x(z^{\bar{i}}) \left[ p_t(x) u_t^i(z^i, x) - p_t(z) u_t^i(x^i, z) \right]$, measuring the total outgoing flux $x \to z$ minus the total incoming flux $z \to x$ for state $x \in \mathcal{S}$. Here $\delta_x(z^{\bar{i}}) = \prod_{j \neq i} \delta_{x^j}(z^j)$, which indicates that we only consider $x$ and $z$ when they only differ in the $i$-th coordinate for calculating the flux [37, 34]. Intuitively, Eq. 3 expresses that the rate of probability at $x$ is equal to the final remaining probability flux $p_t u_t$ at $x$. Previous studies [37, 39] have shown that if the Continuity Equation is satisfied, then $u_t$ is said to generate the probability path $p_t$ as in Definition 1.

# 3 FUDOKI: A Multimodal Model Purely Based on Discrete Flow Matching

This section introduces FUDOKI, a new multimodal architecture that unifies vision and language through the novel lens of discrete flow matching. By adopting this framework, FUDOKI enables an integrated approach to both perception and generation across visual and textual modalities.

## 3.1 Metric-induced Probability Paths with Kinetic Optimal Velocities

Based on the recent theoretical advancement of discrete flow matching [38], we adopt a more general probability path for FUDOKI, instead of the commonly used mask-based mixture paths [37, 36, 35, 46, 45]. Specifically, we consider the probability paths induced by discrete metrics. Given a distance function $d : \mathcal{T} \times \mathcal{T} \to \mathbb{R}_{\geq 0}$ satisfying $d(x^i, x_1^i) = 0$ if and only if $x^i = x_1^i$, we define a path of conditional distributions via:

$$p_t(x^i \mid x_1^i) = \text{softmax}(-\beta_t \cdot d(x^i, x_1^i)), \tag{4}$$

where $\beta_t : [0, 1] \to \mathbb{R}_{\geq 0}$ is a monotonic schedule with boundary values $\beta_0 = 0$, $\beta_1 = \infty$. At $t = 0$, this yields a uniform distribution, and as $t \to 1$, the distribution converges to a delta function at $x_1^i$. Compared to the previous mask-based probability path (*i.e.*, Eq. 1), this metric-induced probability path defines a more semantically meaningful transformation, allowing the probabilities of tokens similar to $x_1^i$ to also increase as $t \to 1$, when setting $d(\cdot, \cdot)$ to measure token embedding distances.

After defining the prescribed metric-induced probability path, we then obtain the probability velocities via minimizing the kinetic energy [38]. In other words, it is expected to minimize the magnitude of flux $p_t u_t$ for probability velocities to obtain a smooth transformation along the probability path. Meanwhile, the obtained velocities should also satisfy several conditions, including the Continuity Equation (*i.e.*, Eq. 3), the non-negativity of the flux between different states (*i.e.*, Eq. 2), and the boundary conditions for $p$ and $q$. We leave the detailed mathematical formulations in the appendix. In this way, the kinetic optimal velocity for Eq. 4 can be formulated as follows [38],

$$u_t^i(x^i, z \mid x_1) = p_t(x^i \mid x_1^i) \dot{\beta}_t [d(z^i, x_1^i) - d(x^i, x_1^i)]_+ \tag{5}$$

where $[\cdot]_+ = \max\{\cdot, 0\}$ is the ReLU operator and $\dot{\beta}_t$ is the derivative of $\beta_t$ w.r.t $t$. Intuitively, for the $i$-th coordinate $z^i \in \mathcal{T}$, this velocity ensures that probability mass flows from state $z^i$ to state $x^i$ only when $x^i$ lies closer to $x_1^i$ than $z^i$ does, *i.e.*, $d(x^i, x_1^i) < d(z^i, x_1^i)$. As a result, the flow monotonically progresses toward $x_1^i$. After introducing the mathematical foundation of discrete flow matching, we now dive into FUDOKI's model structure details.

---

[5] $o(h)$ refers to a function that vanishes at a faster rate than $h$ as $h \to 0$, *i.e.*, $\lim_{h \to 0} \frac{o(h)}{h} = 0$.

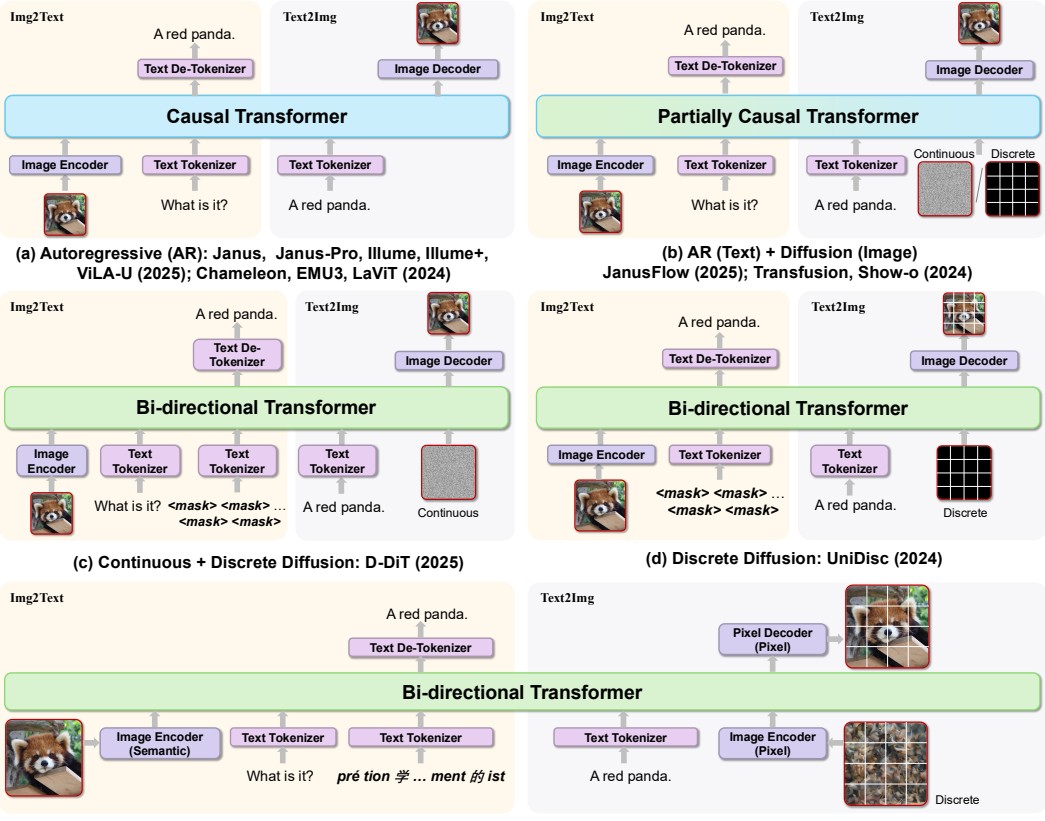

Figure 2: **Comparison of Model Architectures in Unified Multimodal Models.** (a) AR-based models [20, 26, 21, 52–54, 18, 55] perform multimodal tasks via sequential token generation under strictly causal context modeling. (b) Hybrid AR+Diffusion models, such as Transfusion [19] and Show-o [56], integrate AR for text and diffusion models for images, enabling improved visual generation quality. (c-d) Diffusion-based models: D-DiT [46] applies mask-based discrete diffusion to text and continuous diffusion to images, while UniDisc [48] employs mask-based discrete diffusion for both modalities. (e) FUDOKI adopts a unified discrete flow matching framework for both modalities, leveraging a metric-induced probability path to enhance performance in understanding and generation tasks. The inference advantages of FUDOKI over mask-based discrete diffusion modeling used in (c-d) are shown in Fig. 3.

## 3.2 Architecture Overview

As shown in Fig. 2(e), FUDOKI is based on the Janus-1.5B [20] architecture, with minor adaptations to support unified vision-language discrete flow modeling. Specifically, to facilitate effective learning and accelerate convergence, 1) we adopt a full attention mask instead of the standard causal mask to allow all tokens to attend to each other, which helps the model better capture global context; 2) we apply a shifting operation [49] to the output logits by one position, so that our model can inherit the next-token prediction capabilities of AR-based MLLMs as much as possible; 3) unlike continuous diffusion models [57, 12], we do not incorporate additional time embedding layers in the model to explicitly indicate the noise level in the corrupted input. Following the intuition of mask-based discrete diffusion models [49, 58], we observe that our discrete generative model can also implicitly infer the timesteps from the corrupted input along our defined metric-induced probability path (*i.e.*, Eq. 4), resulting in faster adaptation in experiments. The rest of the architecture remains identical to Janus-1.5B. For the text modality, we use the tokenizer with a vocabulary size of $102,400$. For images, we decouple the processing paths for understanding and generation. The semantic encoder SigLIP [59] extracts high-dimensional features for image understanding, which are reshaped and mapped into the LLM input space via an adaptor. For image generation, we follow LlamaGen [60], employing a pixel encoder and decoder to convert images into discrete tokens, with the image token vocabulary size set to $16,384$. Each image token embedding is further transformed into an input

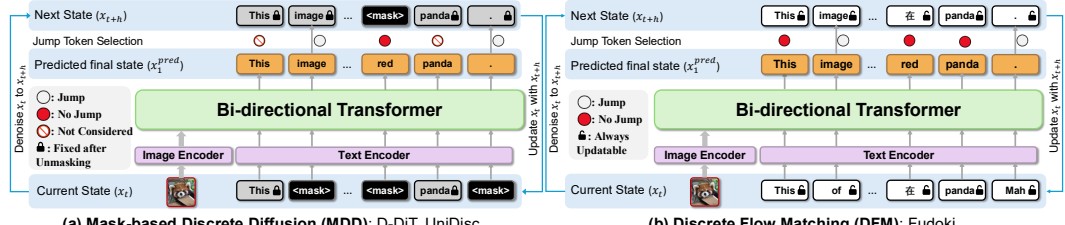

**(a) Mask-based Discrete Diffusion (MDD)**: D-DiT, UniDisc      **(b) Discrete Flow Matching (DFM)**: Fudoki

Figure 3: **Inference Comparisons between (a) Mask-Based Discrete Diffusion Models and (b) Discrete Flow Matching-Based FUDOKI.** In mask-based discrete diffusion models, once a token is unmasked, it typically cannot be modified again, which hinders self-correction. In contrast, our proposed FUDOKI allows its responses to be continuously updated during inference, enabling potential corrections.

feature via a generation adaptor before being fed into the LLM. At the output stage, we use two output heads, a text head and an image head, which convert the transformer outputs into discrete categorical distributions. The appropriate head is selected depending on the target modality during inference. Comparisons with previous AR-based and diffusion-based MLLMs are shown in Fig. 2.

### 3.3 Training

We follow the discrete flow matching framework [34] for model training. Our model is initialized from the pretrained weights of Janus-1.5B [20] and further adapted to our collected dataset, which contains both text-to-image (generation) and image-to-text (understanding) data. Specifically, we divide the training of FUDOKI into two stages: 1) The main goal of the first stage is to quickly relearn the AR-based LLM such that it can effortlessly support the discrete flow matching paradigm. To this end, we only fine-tune the parameters of the transformer while keeping other parts of the model frozen, including the semantic encoders and embedding adaptors. This can help accelerate convergence and stabilize our training; 2) After the first stage, we further fine-tune the whole model to enhance its overall performance on understanding and generation based on discrete flow matching.

Specifically, in each training stage, the ground-truth target $x_1$ is drawn from the data distribution $q(\cdot)$, where the condition is either a text prompt (for T2I) or an image-question pair (for I2T). The target $x_1$ is the image token sequence in the T2I setting and the textual token sequence in the I2T setting. At each training step, a time $t \in [0, 1]$ is uniformly sampled, and a noised sequence $x_t$ is sampled according to the defined probability path $p_t(\cdot \mid x_1)$ in Eq. 4. We set the distance function $d(\cdot, \cdot)$ to measure the L2-distances between normalized token embeddings, which helps increase the probability of sampling tokens whose embeddings are close to the corresponding ground-truth token $x_1^i$ in the embedding space, thereby making the corruption process more semantically meaningful and facilitating learning. The model then receives $x_t$ as input and predicts $x_1$, outputting per-token logits for each position. The training loss is defined as the expected cross-entropy between the ground-truth sequence $x_1$ and the model's predicted distribution:

$$\mathcal{L}_{\text{CE}}(\theta) = \mathbb{E}_{t \sim U[0,1], \, x_1 \sim q(\cdot), \, x_t \sim p_t(\cdot|x_1)} \left[ -\sum_{i=1}^{D} \log p_{1|t}^\theta \left( x_1^i \mid x_t \right) \right] \tag{6}$$

where $p_{1|t}^\theta(\cdot \mid x_t)$ denotes the model's predicted categorical distribution for the $i$-th position, parameterized by $\theta$, given input $x_t$.

### 3.4 Inference

During inference, we apply an Euler solver for more robust sampling as suggested in [38]. This solver simulates the continuous-time Markov chain (CTMC) process $(x_t)_{0 \le t \le 1}$. Given that $x_t \sim p_t$, the solver updates the $i$-th coordinate from time $t$ to $t + h$ using the following procedure:

- Sample $x_1^i \sim p_{1|t}^i(\cdot|x_t)$ from our model;
- Compute the total conditional transition rate $\lambda^i = \sum_{x^i \neq x_t^i} u_t^i(x^i, x_t^i|x_1^i)$ (see Eq. 5);
- Draw a uniform random variable $Z_{\text{change}}^i \sim U[0, 1]$;

- Sample $x_{t+h}^i$ as follows: if $Z_{\text{change}}^i \leq 1 - e^{-h\lambda^i}$, sample $x_{t+h}^i$ from $\frac{u_t^i(\cdot, x_t^i | x_1^i)}{\lambda^i}(1 - \delta_{x_t^i}(\cdot))$; otherwise set $x_{t+h}^i = x_t^i$. Here $\delta_{x_t^i}(\cdot)$ is a delta function.

We provide a detailed understanding of this inference process as follows. In the second step, $\lambda^i$ can be interpreted as the intensity with which the probability mass at $x_t^i$ flows to other states $x^i \neq x_t^i$. The probability that $x_t^i$ will change at the current timestep is determined by comparing the threshold $1 - e^{-h\lambda^i}$ with a uniform random variable $Z_{\text{change}}^i$: the larger $\lambda^i$ is, the more likely a jump will occur. If a change happens, $x_{t+h}^i$ is sampled from all other possible states according to the distribution proportional to $u_t^i(\cdot, x_t^i | x_1^i)$, as defined in Eq. 5. This means the update tends to move $x_{t+h}^i$ towards states that are closer to the model's prediction $x_1^i$. In this way, our sampling process enables the model to: (1) continuously refine its predictions along the probability path, and (2) flexibly adjust tokens towards semantically similar alternatives at each timestep. As shown in Fig. 3, this is in contrast to previous mask-based discrete diffusion models [36, 35, 45], where once a token is unmasked, it generally cannot be modified again, even if it contains an error.

# 4 Experiments

## 4.1 Implementation Details

In both training stages, we use approximately 13M supervised finetuning data to learn our FUDOKI, including 9M in-house generation data for text-to-image generation and 4M public understanding data, which covers various aspects including OCR [61, 62], doc [63], chart [64], screen [65], math [66, 67], language [68], etc. This is less than Chameleon's 1.4B data [54] and LWM's 1B data [69]. We leave the detailed dataset collections in the appendix. For text generation, the sequence length for the response is set to $500$, while for image generation, it is set to $576$ to match the input size of the image encoder. The text embeddings for calculating the metric distance function $d(\cdot, \cdot)$ are taken from the original embedding layer of Janus-Pro-7B [26] and the image embeddings are obtained from the codebook of LlamaGen [60]. We set $\beta_t = c\left(\frac{t}{1-t}\right)^a$ with $c = 3$ and $a = 0.9$, as suggested in [38]. Besides, following previous studies [45, 44], for the text modality, we pad each sequence with `<eos>` (end-of-sequence) and `<pad>` tokens to the maximum length during training, and compute the loss over model's answer tokens, including these special tokens. After the sampling process, we only keep the model responses ahead of the first `<eos>` token. The sampling iterations are set as 32 by default, and the resolution of generated images by FUDOKI is $384 \times 384$. The entire training process spanned approximately 43,000 GPU hours.

## 4.2 Comparison with State-of-the-arts

**Visual Generation Performance**. We evaluate the generation capabilities of FUDOKI on the widely used GenEval benchmark [75]. Table 1 presents the summarized comparisons, where FUDOKI achieved competitive overall performance ($0.77$), matching the top score of prior models in the category of both the generation-only and the understanding-and-generation categories. These results underscore our model's advantages in accurate multi-object understanding and attribute binding, making it promising for complex visual generation tasks that go beyond simple object depiction. This can be attributed to the discrete flow matching framework of FUDOKI, which allows visual information to integrate in both directions for better layout design of generated images.

Besides, we evaluate the visual generation performance of FUDOKI on DPG-Bench [76] (Dense Prompt Graph Benchmark), a comprehensive dataset comprising 1,065 lengthy and densely composed prompts specifically designed to assess the fine-grained semantic alignment capabilities of text-to-image models. As shown in Table 2, FUDOKI demonstrates competitive performance compared to both generation-specialized and unified multimodal models. These results highlight FUDOKI's strong ability to handle complex, information-rich prompts, establishing it as a robust and versatile solution for multi-aspect visual generation tasks.

**Multimodal Understanding**. We evaluate the understanding capabilities of FUDOKI on several benchmarks, including POPE [91], MME-P [92], SEED [93], MMB [94], GQA [95], MMMU [96],

Table 1: **Visual Generation Performance on the GenEval Benchmark**. "*Und.*" and "*Gen.*" denotes "*Understanding*" and "*Generation*". [†] denotes models that integrate an external pretrained diffusion model.

| Type | Paradigm | Method | Single Obj. | Two Obj. | Counting | Colors | Position | Color Attri. | Overall↑ |
|---|---|---|---|---|---|---|---|---|---|
| *Gen. Only* | AR | LlamaGen [60] | 0.71 | 0.34 | 0.21 | 0.58 | 0.07 | 0.04 | 0.32 |
| | | Emu3-Gen [18] | 0.98 | 0.71 | 0.34 | 0.81 | 0.17 | 0.21 | 0.54 |
| | Diffusion | LDM [12] | 0.92 | 0.29 | 0.23 | 0.70 | 0.02 | 0.05 | 0.37 |
| | | SDv1.5 [12] | 0.97 | 0.38 | 0.35 | 0.76 | 0.04 | 0.06 | 0.43 |
| | | PixArt-$\alpha$ [13] | 0.98 | 0.50 | 0.44 | 0.80 | 0.08 | 0.07 | 0.48 |
| | | SDv2.1 [12] | 0.98 | 0.51 | 0.44 | 0.85 | 0.07 | 0.17 | 0.50 |
| | | DALL-E 2 [70] | 0.94 | 0.66 | 0.49 | 0.77 | 0.10 | 0.19 | 0.52 |
| | | SDXL [71] | 0.98 | 0.74 | 0.39 | 0.85 | 0.15 | 0.23 | 0.55 |
| | | DALL-E 3 [72] | 0.96 | 0.87 | 0.47 | 0.83 | 0.43 | 0.45 | 0.67 |
| | | SD3-Medium [14] | 0.99 | 0.94 | 0.72 | 0.89 | 0.33 | 0.60 | 0.74 |
| *Und. and Gen.* | AR | SEED-X[†] [73] | 0.97 | 0.58 | 0.26 | 0.80 | 0.19 | 0.14 | 0.49 |
| | | LWM [69] | 0.93 | 0.41 | 0.46 | 0.79 | 0.09 | 0.15 | 0.47 |
| | | ILLUME [21] | 0.99 | 0.86 | 0.45 | 0.71 | 0.39 | 0.28 | 0.61 |
| | | TokenFlow-XL [74] | 0.95 | 0.60 | 0.41 | 0.81 | 0.16 | 0.24 | 0.55 |
| | | Chameleon [54] | - | - | - | - | - | - | 0.39 |
| | | Janus [20] | 0.97 | 0.68 | 0.30 | 0.84 | 0.46 | 0.42 | 0.61 |
| | | Janus-Pro-1B [26] | 0.98 | 0.82 | 0.51 | 0.89 | 0.65 | 0.56 | 0.73 |
| | AR+Diffusion | Show-o [56] | 0.95 | 0.52 | 0.49 | 0.82 | 0.11 | 0.28 | 0.53 |
| | | Transfusion [19] | - | - | - | - | - | - | 0.63 |
| | Diffusion | UniDisc [48] | 0.92 | 0.47 | 0.15 | 0.67 | 0.13 | 0.19 | 0.42 |
| | | D-DiT [46] | 0.97 | 0.80 | 0.54 | 0.76 | 0.32 | 0.50 | 0.65 |
| | Discrete Flow | **FUDOKI (Ours)** | 0.96 | 0.85 | 0.56 | 0.88 | 0.68 | 0.67 | 0.77 |
| | | **+Inference Scaling** | 0.98 | 0.95 | 0.73 | 0.94 | 0.88 | 0.78 | 0.88 |

Table 2: **Visual Generation Performance on DPG-Bench.**

| Method | Global | Entity | Attribute | Relation | Other | Overall↑ |
|---|---|---|---|---|---|---|
| SDv1.5 [12] | 74.63 | 74.23 | 75.39 | 73.49 | 67.81 | 63.18 |
| PixArt-$\alpha$ [13] | 74.97 | 79.32 | 78.60 | 82.57 | 76.96 | 71.11 |
| Lumina-Next [77] | 82.82 | 88.65 | 86.44 | 80.53 | 81.82 | 74.63 |
| SDXL [71] | 83.27 | 82.43 | 80.91 | 86.76 | 80.41 | 74.65 |
| Playground v2.5 [78] | 83.06 | 82.59 | 81.20 | 84.08 | 83.50 | 75.47 |
| Hunyuan-DiT [79] | 84.59 | 80.59 | 88.01 | 74.36 | 86.41 | 78.87 |
| PixArt-$\Sigma$ [80] | 86.89 | 82.89 | 88.94 | 86.59 | 87.68 | 80.54 |
| Emu3-Gen [18] | 85.21 | 86.68 | 86.84 | 90.22 | 83.15 | 80.60 |
| DALL-E 3 [72] | 90.97 | 89.61 | 88.39 | 90.58 | 89.83 | 83.50 |
| SD3-Medium [14] | 87.90 | 91.01 | 88.83 | 80.70 | 88.68 | 84.08 |
| Janus [20] | 82.33 | 87.38 | 87.70 | 85.46 | 86.41 | 79.68 |
| Janus-Pro-1B [26] | 87.58 | 88.63 | 88.17 | 88.98 | 88.30 | 82.63 |
| **FUDOKI (Ours)** | 80.55 | 89.73 | 88.05 | 93.66 | 78.00 | 83.63 |

and MM-Vet [97]. Table 3 presents the summarized results [6]. Notably, our FUDOKI model (1.5B parameters) achieved highly competitive results, which are on par with or surpass several AR-based MLLMs of similar or even larger scale. This demonstrates that FUDOKI delivered robust multimodal understanding capabilities, which can be attributed to the bidirectional reasoning property of discrete flow matching. Moreover, we provide generation process comparisons for understanding in Fig. 4, which further highlight the advantages of sampling through discrete flow matching for reasoning, *e.g.*, self-correcting the reasoning process for coherency. Our findings highlight the effectiveness and efficiency of FUDOKI, making it a strong alternative to the established AR-based MLLMs.

**Inference Scaling**. We applied test-time inference scaling techniques [50] to FUDOKI, leveraging a judge model to score multiple candidate outputs and select the highest-scoring responses. The last rows of Table 1 and Table 3 illustrate the impact of inference scaling on visual generation and understanding. For generation, we used the VILA-Judge model [98] to select the top 4 images from 32 candidates per prompt in the GenEval benchmark, resulting in significant performance gains. For understanding, we employed an LLM as the judge to choose the best response from 8 candidates in the challenging MMVet benchmark, where improvements were observed. These results highlight FUDOKI's potential for further enhancement through reinforcement learning approaches [1, 99].

---

[6]UniDisc [48] is not included in the table due to their inability to conduct visual question answering tasks.

Table 3: **Multimodal Understanding Performance on Various Benchmarks.** "*Und.*" and "*Gen.*" denotes "*Understanding*" and "*Generation*". [†] denotes models that integrate an external pretrained diffusion model.

| Type | Paradigm | Model | # LLM Params | POPE↑ | MME-P↑ | MMB↑ | SEED↑ | GQA↑ | MMMU↑ | MM-Vet↑ |
|------|----------|-------|--------------|-------|--------|------|-------|------|-------|---------|
| *Und. Only* | AR | LLaVA-v1.5-Phi-1.5 [56] | 1.3B | 84.1 | 1128.0 | - | - | 56.5 | 30.7 | - |
| | | MobileVLM [81] | 1.4B | 84.5 | 1196.2 | 53.2 | - | 56.1 | - | - |
| | | MobileVLM-V2 [82] | 1.4B | 84.3 | 1302.8 | 57.7 | - | 59.3 | - | - |
| | | MobileVLM [81] | 2.7B | 84.9 | 1288.9 | 59.6 | - | 59.0 | - | - |
| | | MobileVLM-V2 [82] | 2.7B | 84.7 | 1440.5 | 63.2 | - | 61.1 | - | - |
| | | LLaVA-Phi [83] | 2.7B | 85.0 | 1335.1 | 59.8 | - | - | - | 28.9 |
| | | LLaVA [6] | 7B | 76.3 | 809.6 | 38.7 | 33.5 | - | - | 25.5 |
| | | LLaVA-v1.5 [84] | 7B | 85.9 | 1510.7 | 64.3 | 58.6 | 62.0 | 35.4 | 31.1 |
| | | InstructBLIP [8] | 7B | - | - | 36.0 | 53.4 | 49.2 | - | 26.2 |
| | | Qwen-VL-Chat [85] | 7B | - | 1487.5 | 60.6 | 58.2 | 57.5 | - | - |
| | | IDEFICS-9B [86] | 8B | - | - | 48.2 | - | 38.4 | - | - |
| | | Emu3-Chat [18] | 8B | 85.2 | 1244 | 58.5 | 68.2 | 60.3 | 31.6 | 37.2 |
| | | InstructBLIP [8] | 13B | 78.9 | 1212.8 | - | - | 49.5 | - | 25.6 |
| *Und. and Gen.* | AR | LaVIT[†] [87] | 7B | - | - | - | - | 46.8 | - | - |
| | | MetaMorph[†] [88] | 8B | - | - | 75.2 | 71.8 | - | - | - |
| | | Gemini-Nano-1 [89] | 1.8B | - | - | - | - | - | 26.3 | - |
| | | ILLUME [21] | 7B | 88.5 | 1445.3 | 65.1 | 72.9 | - | 38.2 | 37.0 |
| | | TokenFlow-XL [74] | 13B | 86.8 | 1545.9 | 68.9 | 68.7 | 62.7 | 38.7 | 40.7 |
| | | LWM [69] | 7B | 75.2 | - | - | - | 44.8 | - | 9.6 |
| | | VILA-U [90] | 7B | 85.8 | 1401.8 | - | 59.0 | 60.8 | - | 33.5 |
| | | Chameleon [54] | 7B | - | - | - | - | - | 22.4 | 8.3 |
| | | Janus [20] | 1.5B | 87.0 | 1338.0 | 69.4 | 63.7 | 59.1 | 30.5 | 34.3 |
| | | Janus-Pro-1B [26] | 1.5B | 86.2 | 1444.0 | 75.5 | 68.3 | 59.3 | 36.3 | 39.8 |
| | AR+Diffusion | Show-o-256 [56] | 1.3B | 73.8 | 948.4 | - | - | 48.7 | 25.1 | - |
| | | Show-o-512 [56] | 1.3B | 80.0 | 1097.2 | - | - | 58.0 | 26.7 | - |
| | Diffusion | D-Dit [46] | 2.0B | 84.0 | 1124.7 | - | - | 59.2 | - | - |
| | Discrete Flow | **FUDOKI (Ours)** | 1.5B | 86.1 | 1485.4 | 73.9 | 68.2 | 57.6 | 34.3 | 38.0 |
| | | **+Inference Scaling** | 1.5B | - | - | - | - | - | - | 55.5 |

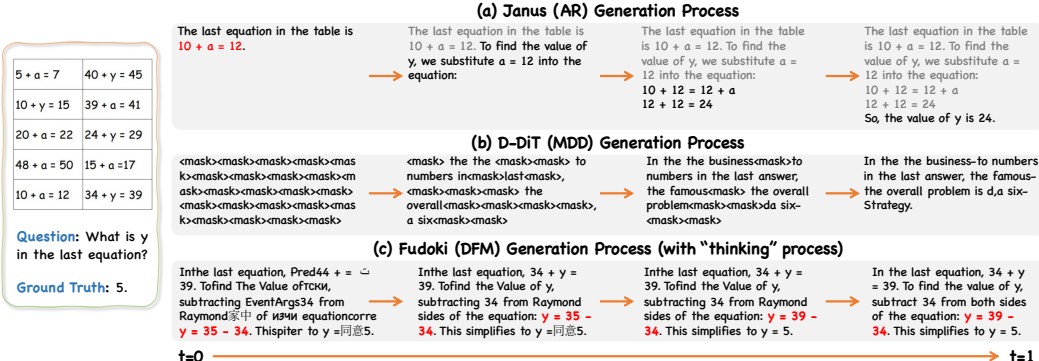

Figure 4: **Generation process of different methods.** (a) AR-based Janus can only generate tokens sequentially; if an error is made in the initial step, subsequent outputs will consistently propagate this mistake. (b) D-DiT (mask-based discrete diffusion, MDD) cannot revise tokens once unmasked, making errors irreversible and leading to poor generalization. (c) FUDOKI (discrete flow matching, DFM) allows generated tokens to be revised in subsequent steps, enabling step-by-step reasoning and error correction for more accurate answers.

## 4.3 Ablation Studies

**Training Strategies.** 1) *AR Initialization vs Training from Scratch*: As shown in Fig. 5 (left), we compare models initialized with autoregressive (AR) weights [20] against models trained from scratch. The results indicate that AR initialization provided a substantial advantage for accelerating model training, leading to consistently lower training loss throughout the optimization process. 2) *Effects of Time-embedding Layers*: We also evaluate the impact of incorporating time embedding layers into the model architecture. The results in Fig. 5 (middle) show that the model without time embedding layers consistently achieves slightly lower training loss than the version with time embeddings. This

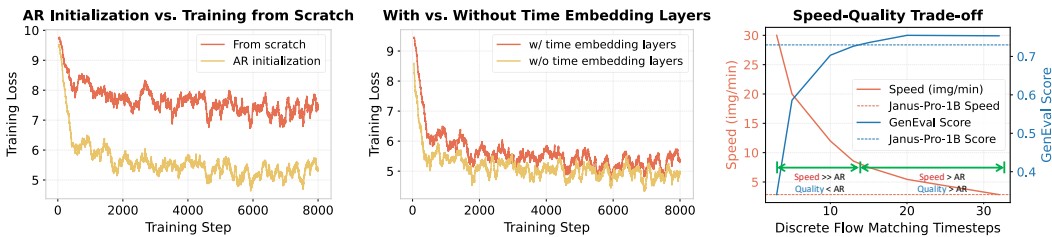

Figure 5: **Comparison of training loss and speed-quality trade-off. (Left, Middle)** AR initialization and removing time embedding layers both reduce training loss. **(Right)** With fewer timesteps, FUDOKI achieves much higher speed but slightly lower quality than AR; at the optimal timestep, both metrics surpass the AR.

Table 4: **Quantitative comparisons between the AR-based models and our proposed FUDOKI in terms of the self-correcting capabilities.**

| Method | Baseline | +Janus-Pro-1B to correct | +Janus-Pro-7B to correct | +FUDOKI to correct |
|--------|----------|--------------------------|--------------------------|--------------------|
| **MMVet** | 37.98 | 36.33 (-1.65) | 38.30 (+0.32) | 38.53 (+0.55) |

suggests that our discrete generative model can implicitly infer timesteps from corrupted input, and removing time embeddings reduces model complexity.

**Quality-Speed Trade-off.** Fig. 5 (right) illustrates the trade-off between speed (in images per minute) and quality (GenEval score) in terms of setting different inference timesteps for visual generation. It compares the inference performance of FUDOKI with the autoregressive (AR) baseline, Janus-Pro-1B (with KV cache enabled). The red solid line (to the left vertical axis) represents the speed of FUDOKI, which decreases as the number of timesteps increases, while the blue solid line (to the right vertical axis) represents the generation quality of FUDOKI, which improves and stabilizes as timesteps increase. We also draw the dashed horizontal lines indicating the baseline values for Janus-Pro-1B, with the red dashed line for speed and the blue dashed line for quality. Please pay attention to the intersection point of the green arrows. This intersection marks the point where FUDOKI achieves a significant speed advantage over the AR baseline (as the red solid line exceeds the red dashed line) and comparable output quality (where the blue solid line meets the blue dashed line). This can be attributed to FUDOKI's fewer inference steps and richer bidirectional context modeling.

**Results on the Self-Correction Capability**. We quantitatively evaluated the self-correcting capabilities of FUDOKI and performed comparisons with the AR-based models. In experiments, both FUDOKI and AR-based models were tasked with correcting baseline responses where necessary. The baseline responses were obtained from Janus-Pro-1B on the MMVet benchmark, using the Open-Compass VLMEvalKit codebase [100]. To assess their correction abilities: 1) For AR-based models, we appended the following prompt to the original prompt: "Your original response is: <placeholder>. Please correct it if needed. Otherwise, you may keep it the same." The models were then evaluated on their ability to revise or retain the response as appropriate; 2) For FUDOKI, we initialized the responses with the baseline responses (rather than uniformly-sampled noise tokens) and performed iterative refinements over 32 steps, as described in the paper. As shown in Table 4, FUDOKI achieved the highest performance improvement, while Janus-Pro-1B's performance declined and Janus-Pro-7B showed less increase, despite its larger model size than ours. We attribute such results to the increased context length introduced by the baseline responses, which may distract the AR-based model's focus. This further highlights the limitations of the AR paradigm for effective self-correction.

## 5 Conclusion

In this work, we introduced FUDOKI, a multimodal model that uses discrete flow matching to unify visual understanding and generation. Unlike conventional autoregressive and masking-based approaches, FUDOKI leverages discrete flow matching for iterative self-correction, bidirectional reasoning, and flexible generation. Experiments show that FUDOKI performs competitively with leading AR-based MLLMs on both visual understanding and text-to-image generation tasks. These results highlight discrete generative flow models—exemplified by FUDOKI—as a promising direction for advancing multimodal language models and meeting future AGI challenges.

## Acknowledgments

This paper is partially supported by the National Key R&D Program of China No.2022ZD0161000 and the General Research Fund of Hong Kong No.17208825 and 17209324.

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

# A  Related Work

## A.1  Unified multimodal LLMs

**Autoregressive Paradigms: End-to-End and Two-Stage Modeling.** Autoregressive (AR) modeling remains a core strategy for unified multimodal understanding and generation, but recent advances have led to two distinct AR-based paradigms.

The first is the *end-to-end AR paradigm*, in which all modalities—including images, text, video, and even audio—are tokenized into a unified discrete space and directly modeled within a single AR sequence framework. Representative works such as Unified-IO [101, 102], Chameleon [54], AnyGPT [103], and Emu3 [18] follow this approach: a transformer autoregressively predicts the next token across modalities, with image tokens directly decoded back to pixels via learned decoders such as VQGAN. DDT-Llama [104] further improves tokenization by introducing recursive diffusion timestep tokens, enabling better alignment with language modeling and image reconstruction. This approach enables strong performance in both understanding and generation, and supports flexible modality conversion (e.g., AnyGPT covers speech and music). Building on this foundation, models like Janus [20] and Janus-Pro [26] decouple visual encoding for understanding and generation to address the granularity mismatch, while VILA-U [90], LWM [69], and LaVIT [55] focus on efficient tokenization, unified visual-text alignment, and scaling to long-context and video scenarios. Illume [21] and Illume+ [52] further enhance data efficiency and token alignment, with Illume+ introducing dual visual tokenization and a diffusion-based decoder for higher-fidelity image synthesis and editing.

By contrast, the *two-stage AR+diffusion paradigm* separates sequence modeling and image synthesis: AR models first generate image tokens, which are then used as conditions for downstream diffusion decoders to boost image quality and diversity. Representative works include DreamLLM [105], which enables free-form interleaved multimodal generation; MiniGPT-5 [106], which improves image-text coherence with a two-stage pipeline; NExT-GPT [107], which supports any-to-any modality conversion by connecting AR sequence modeling with modular diffusion decoders; MetaMorph [88], which efficiently adapts LLMs for unified text and visual token generation; SEED-LLaMA [17], which aligns image token semantics with text for scalable multimodal autoregression; and SEED-X [73], which further enables arbitrary-size and multi-granularity image generation. Recently, BLIP3-o [108] advanced this paradigm by generating CLIP-based image features using a diffusion transformer and adopting sequential pretraining to better balance understanding and generation. Collectively, these models demonstrate the flexibility and high image fidelity achievable with the two-stage approach, highlighting a distinct trade-off with end-to-end AR models in reasoning and generation quality.

**Hybrid Paradigm: Integrating AR and Diffusion within a Unified Framework.** To bridge the gap between the reasoning strengths of AR models and the generative power of diffusion models, hybrid paradigms have emerged that combine both mechanisms in a unified architecture. For example, JanusFlow [109] employs a continuous reactified flow for image generation, Show-o [56] adopts a discrete MaskGIT-style diffusion, while Transfusion [19] utilizes a continuous U-Net-based DDPM. Despite their differences in diffusion implementation, these hybrid models all enable more flexible and controllable vision-language generation, further blurring the boundaries between AR and diffusion approaches.

**Diffusion Paradigm: Fully Diffusion-Based Multimodal Generation.** In parallel, fully diffusion-based approaches have also been proposed for unified multimodal modeling. UniDisc [48] and D-Dit [46] formulate both text and image generation as a discrete diffusion process, starting from masked sequences and enabling joint inpainting, editing, and controllable multimodal generation. By leveraging the iterative denoising process, diffusion models typically achieve superior generation fidelity and support fine-grained, high-quality editing. Moreover, unlike autoregressive models that generate tokens sequentially, diffusion-based approaches can produce multiple tokens in parallel during inference, improving efficiency and enabling more globally consistent outputs. While these models offer enhanced controllability and flexible inference, they may still face challenges in complex instruction following and sequential reasoning. Nevertheless, fully diffusion-based paradigms represent a promising direction for scenarios requiring fine-grained editing, state-of-the-art generation quality, and efficient parallel decoding across modalities.

**Comparisons with Bagel [25]**. Bagel [25] is a very strong recent advance in unified multimodal understanding and generation. While both FUDOKI and Bagel aim for unified multimodal mod-

eling, they are based on fundamentally different generative paradigms and architectural choices. Specifically, Bagel employs a large Mixture-of-Transformer-Experts (MoT) architecture and follows the autoregressive (AR) modeling paradigm, enabling it to efficiently scale with massive, carefully structured interleaved multimodal data. In contrast, FUDOKI is the first general-purpose unified multimodal model built entirely on discrete flow matching, which allows for bidirectional information integration and iterative self-correction during generation. In terms of empirical performance, Bagel demonstrates strong results on both multimodal generation and understanding, including advanced tasks such as free-form image manipulation. We acknowledge that FUDOKI currently lags behind Bagel, which can be attributed mainly to Bagel's novel data scaling strategies and substantially larger model size (14B parameters for Bagel vs. 1.5B for FUDOKI). We will explore integrating similar scaling approaches in future work.

## A.2 Flow Matching

Flow matching offers a fundamentally different approach to generative modeling compared to diffusion models. While diffusion models rely on repeatedly injecting random noise into data and then iteratively denoising it, flow matching instead learns a smooth, continuous transformation, formulated through ordinary differential equations (ODEs), that maps a simple distribution (such as Gaussian noise) directly to real data. This approach eliminates the need for repeated noise addition and removal.

Pioneering this direction, Lipman et al. [42] introduced Continuous Normalizing Flows (CNFs) and the flow matching framework, which trains neural networks by regressing vector fields along flexible probability paths. This work laid the foundation for subsequent advances in CNF-based generative modeling. Building on this, Liu et al. [41] proposed Rectified Flow, which learns neural ODEs along straight-line paths between distributions, enabling more efficient and scalable training for tasks such as image generation and domain adaptation. More recently, Albergo and Vanden-Eijnden [110] presented InterFlow, which simplifies training by directly inferring the velocity field from the probability flow of an interpolant density, thus avoiding costly ODE backpropagation and supporting efficient likelihood estimation and high-resolution generation.

A key advantage of flow matching is its **sampling efficiency**: by allowing deterministic sampling in just a few ODE steps, it achieves competitive FID scores with orders of magnitude fewer steps compared to diffusion-based samplers. This remarkable efficiency has quickly made flow matching a dominant approach in state-of-the-art image and video generation models.

Recent studies have also extended flow matching to discrete data domains. Campbell et al. [39] introduced Discrete Flow Models (DFMs), which generalize flow matching to discrete spaces using continuous-time Markov chains, improving multimodal modeling of both continuous and discrete data over discrete diffusion models. Similarly, Gat et al. [37] proposed Discrete Flow Matching, a framework that supports general probability paths and scalable non-autoregressive generation, significantly narrowing the performance gap between discrete flow and autoregressive models on coding benchmarks.

Thanks to these advances, flow matching methods have demonstrated strong performance across a wide range of domains, including image synthesis [14, 15], video generation [111–114], speech and audio generation [115–117], protein design [118–120], and robot control [121]. These successes underscore the broad applicability and effectiveness of flow matching frameworks.

## A.3 Discrete Diffusion Models

Diffusion models have achieved remarkable success in continuous domains such as images and audio [57, 122, 123]. However, their adaptation to natural language poses unique challenges due to the discrete nature of text. Early attempts to overcome this primarily injected Gaussian noise into token embedding spaces, followed by denoising to reconstruct discrete sequences [124, 125]. Representative models in this line include Diffusion-LM [124], DiffuSeq [125], and Plaid [126]. While these approaches show promise for controllable generation and sequence-to-sequence tasks, the need to map between discrete and continuous representations complicates training and inference.

Recent research has shifted to discrete noise-based diffusion models to address these limitations, where noise injection and denoising are directly defined in the symbol space. The most influential

early works in this direction are Argmax Flows [127] and D3PM [33]. D3PM, in particular, provides a systematic framework for discrete diffusion, formalizing both absorbing (mask-based) and uniform (categorical) noise processes for sequence corruption. These foundational studies enable the progressive corruption of discrete sequences through distinct forward processes: in the absorbing (mask-based) process, tokens in the original sequence are gradually replaced with a special absorbing token (e.g., <MASK>); in the uniform (categorical) process, tokens are progressively replaced with randomly sampled tokens from the vocabulary. The diffusion model is then trained to reverse these processes, denoising the corrupted sequence back to the original data. Building on these foundations, subsequent models such as DiffusionBERT [58], LLaDA [44], and MD4 [35] introduce improvements in noise scheduling, scalability, and training objectives. Methods like MaskGIT [128] and FiLM [129], although originally proposed for vision or general infilling tasks, are methodologically aligned with mask-based diffusion, employing iterative generation with absorbing masks. These models have achieved performance competitive with, or even superior to, autoregressive models in language modeling, infilling, and reasoning tasks.

In addition to mask-based approaches, the uniform (categorical) transition process, also formalized in D3PM, corrupts sequences by progressively replacing tokens in the original data with tokens sampled uniformly from the vocabulary, rather than a single mask token. SEDD [34] extends score matching to discrete data via a score entropy loss, achieving state-of-the-art results and in some cases surpassing autoregressive baselines. RDM [130] introduces a reparameterized sampling framework to improve training and sampling efficiency. Furthermore, recent studies [131, 132] model discrete diffusion as a continuous-time Markov chain, advancing theoretical understanding and practical efficiency. Most recently, Discrete Flow Matching (DFM) [37] was proposed as a novel discrete flow paradigm for generative modeling of high-dimensional discrete data. Unlike flow matching and diffusion models designed for continuous domains, DFM introduces a general family of probability paths that interpolate between source and target distributions in discrete space, and provides a unified formula for sampling from these paths using learned posteriors such as probability denoisers and noise predictors. Empirically, DFM demonstrates that adopting a uniform (categorical) transition process, rather than an absorbing (mask-based) process, consistently leads to improved generative performance.

Recent scaling studies further demonstrate that, in addition to matching autoregressive models in perplexity and generation quality, discrete diffusion models have achieved strong performance on complex reasoning and planning tasks, underscoring their flexibility and potential as competitive alternatives for natural language generation and understanding [133–136, 44, 35]. Recent work [49] explores directly adapting pretrained autoregressive language models into non-autoregressive diffusion models via continual finetuning, enabling efficient knowledge transfer between paradigms. Building on this line, Dream 7B [45] further advances diffusion LMs by consistently outperforming previous diffusion models and matching the performance of top autoregressive models of similar size.

## B More Comparison with State-of-the-arts

**Qualitative Comparisons on Visual Generation.** Figure 6 presents qualitative comparisons of visual generation results produced by three models: Janus [20], D-DiT [46], and our method, FUDOKI, across a diverse set of text prompts. Each row corresponds to a different prompt, covering scenarios such as animals in unusual environments, cartoon avatars, and objects with specific attributes. As shown in the figure, FUDOKI consistently produced images that more accurately captured the semantics of the prompts, demonstrating superior text-image alignment and higher visual fidelity.

**Qualitative Comparisons on Visual Understanding.** Figure 7 presents qualitative comparisons of visual understanding capabilities among Janus (AR) [20], D-DiT (mask-based discrete diffusion, MDD) [46], and our FUDOKI (discrete flow matching, DFM). The upper section shows selected intermediate outputs from each model's answer generation process, illustrating their reasoning dynamics. The lower section presents additional visual question answering cases, where FUDOKI demonstrates higher reasoning accuracy and better alignment with ground truth answers, highlighting its superior ability to generate reliable and precise responses.

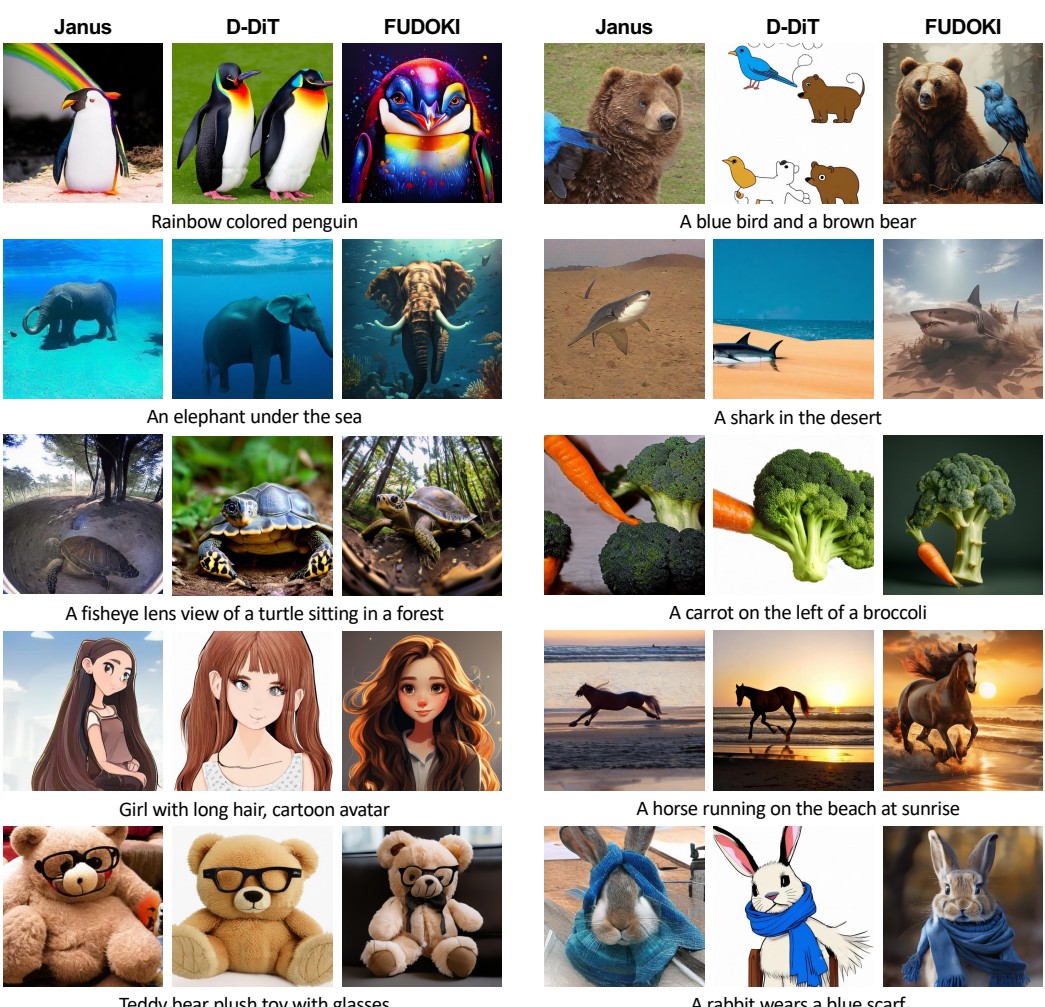

| Janus | D-DiT | FUDOKI | | Janus | D-DiT | FUDOKI |
|-------|-------|--------|---|-------|-------|--------|

Rainbow colored penguin

A blue bird and a brown bear

An elephant under the sea

A shark in the desert

A fisheye lens view of a turtle sitting in a forest

A carrot on the left of a broccoli

Girl with long hair, cartoon avatar

A horse running on the beach at sunrise

Teddy bear plush toy with glasses

A rabbit wears a blue scarf

Figure 6: **Qualitative Comparisons on Visual Generation.** Comparison among Janus [20], D-DiT [46] and FUDOKI on various text prompts. The results demonstrate that our method (FUDOKI) achieved superior text-image alignment and aesthetics.

## C  Further Results

**The Denoising Process of FUDOKI.** Fig. 8 illustrates the iterative refinement process enabled by the discrete flow matching framework in FUDOKI, demonstrating its application to both generation and understanding tasks. The top panel visualizes how images are progressively denoised over iterations, transitioning smoothly from an initial noisy prior $x_0$ to the final high-fidelity image $x_1$. Across diverse generation examples—ranging from animals to objects—the model incrementally sharpens semantic details and corrects spatial structure at each refinement step. The bottom panel depicts a similar iterative refinement for the understanding task, where the model extracts text from an image. Starting from a noisy token sequence, irrelevant or incorrect tokens are gradually replaced with accurate tokens (e.g., "Sara Lee") as the model converges to the correct answer. The red arrows highlight token-level updates during each step, emphasizing the model's ability to systematically and continuously correct errors and align predictions. This figure showcases how discrete flow matching enables fine-grained control and progressive improvement in both modalities by modeling transitions in discrete space, leading to more accurate and coherent outputs. More cases can be found in our project page: fudoki-dfm.github.io/fudoki/.

**Maze Navigation.** In this section, we train our proposed FUDOKI model on a novel task—maze navigation—which simultaneously requires understanding and generation capabilities. To this end,

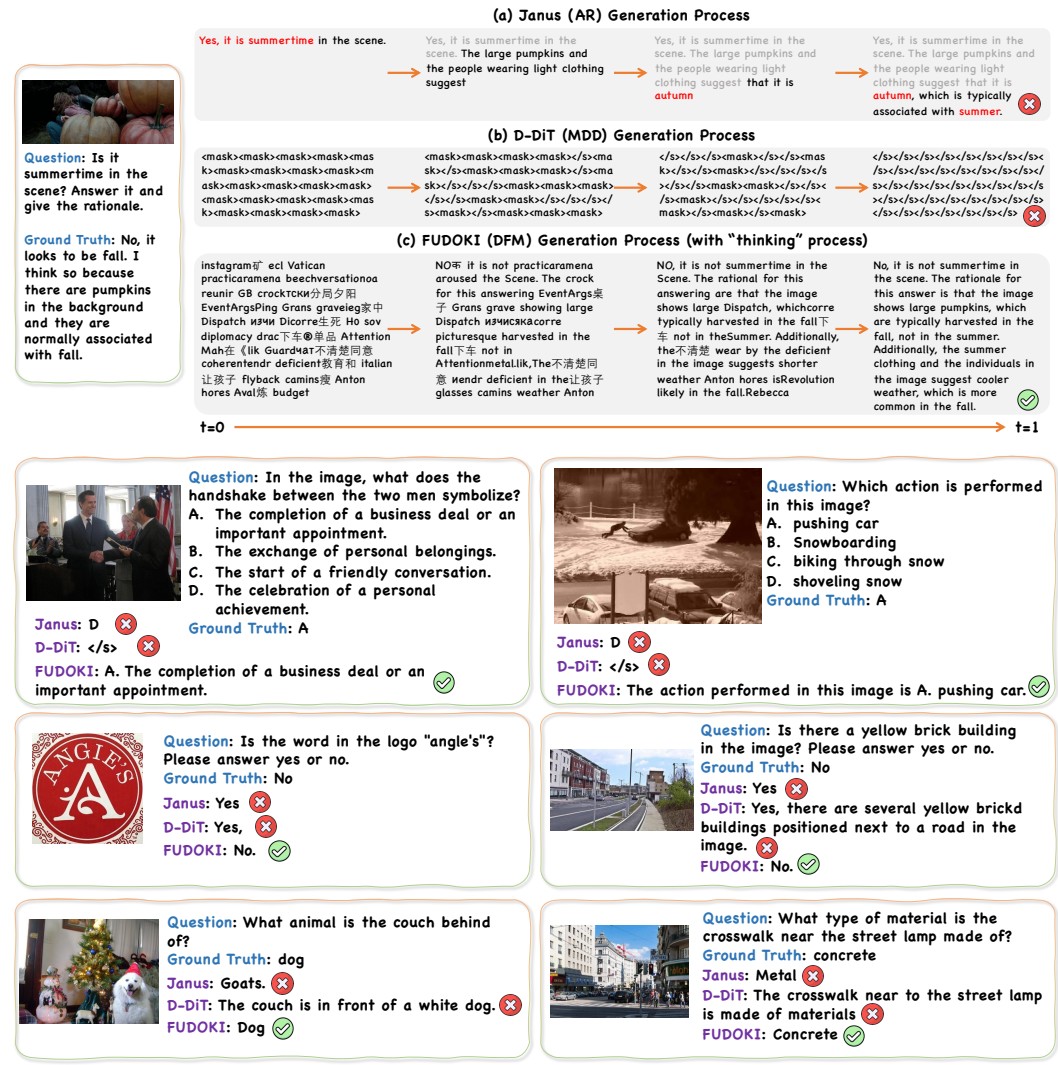

Figure 7: **Qualitative Comparisons on Visual Understanding.** The upper part of the figure shows selected intermediate outputs from the answer generation process of different models—Janus (AR), D-DiT (mask-based discrete diffusion, MDD), and our FUDOKI (discrete flow matching, DFM)—to illustrate their reasoning approaches. Specifically, Janus, the AR-based model, is unable to revise its initial incorrect response (*i.e.*, *"Yes, it is summertime ..."*), even after generating the correct rationale later (*i.e.*, *"The large pumpkins ... suggest that it is autumn"*), making its response inconsistent overall. Meanwhile, D-DiT, the mask-based diffusion model, fails to handle this reasoning task, often producing empty outputs (*i.e.*, only ** tokens). In contrast, our discrete flow matching model, FUDOKI, demonstrates a coherent and accurate reasoning trajectory, producing consistent and correct answers. The lower part of the figure provides additional qualitative examples on visual question answering tasks. FUDOKI consistently delivers more accurate and well-aligned reasoning with the ground truth.

Fig. 9 presents a series of multimodal decision-making scenarios where FUDOKI and GPT-4o/GPT-Image-1 are evaluated on their ability to reason over spatial layouts and produce both textual and visual outputs. Each case involves a frozen lake grid of increasing size ($3\times3$, $4\times4$, and $5\times5$), with a defined goal and a character's current position. The task is to select a safe move that avoids hazards (dark blue holes) while progressing toward the treasure. We notice that while GPT-4o provided well-reasoned textual explanations that include safety considerations, goal alignment, and environmental awareness, its visual updates lacked consistency with its textual responses, and even altered the maze structure (in the third row of the figure). In contrast, FUDOKI consistently predicted plausible directions and generated coherent visual updates aligned with the task constraints, showing basic

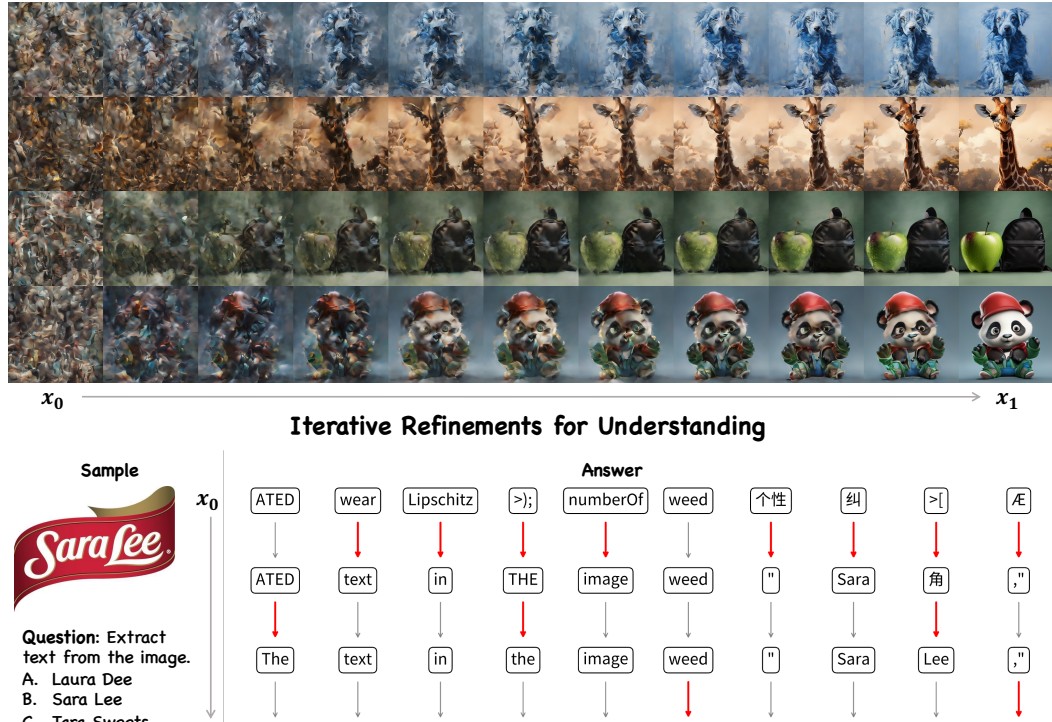

Figure 8: **Visualization of the iterative refinement process enabled by discrete flow matching in FUDOKI**, demonstrating denoising process for text-to-image generation and visual understanding tasks.

Table 5: **Performance Comparisons on the MathVista Benchmark.**

| Method | Janus-1.5B | Janus-Pro-1B | FUDOKI |
|---|---|---|---|
| **MathVista** | 32.4 | 35.1 | 38.6 |

spatial awareness. Furthermore, as shown in Fig. 10, FUDOKI is capable of completing the entire maze navigation sequence, moving from the initial position to the treasure step by step.

**Results on the MathVista [137] Benchmark**. We also evaluated our proposed FUDOKI on a more challenging mathematical reasoning benchmark, MathVista (testmini) [137]. As shown in Table 5, we find that FUDOKI achieved the best performance compared to AR-based models at the same scale. We attribute this improvement to FUDOKI's discrete flow matching framework, which leverages bidirectional context modeling to facilitate complex reasoning.

## D    Dataset Collections

Our training set comprises a total of 12.62 million samples, divided into two main categories: Generation (8.76M, 69%) and Understanding (3.86M, 31%), as shown in Fig. 11. The Generation subset, which is entirely composed of in-house data, is constructed for text-to-image generation tasks. In contrast, the Understanding subset covers a diverse set of information extraction and comprehension tasks. This balanced and large-scale collection ensures comprehensive support for both generative and understanding capabilities.

Specifically, the public Understanding of data covers the following aspects:

- **General** (1506.8K, 40.6%): ShareGPT-4o (57.2K) [138], VSR (12.8K) [139], ALLaVA-Instruct (680.4K) [140], IconQA (29.9K) [141], LVIS-Instruct4V (10.0K) [142], ShareGPT4V (613.3K) [143], VIQuAE (18.5K) [144], RAVEN (0.3K) [145], Visual7W (14.4K) [146], In-house (70.0K)

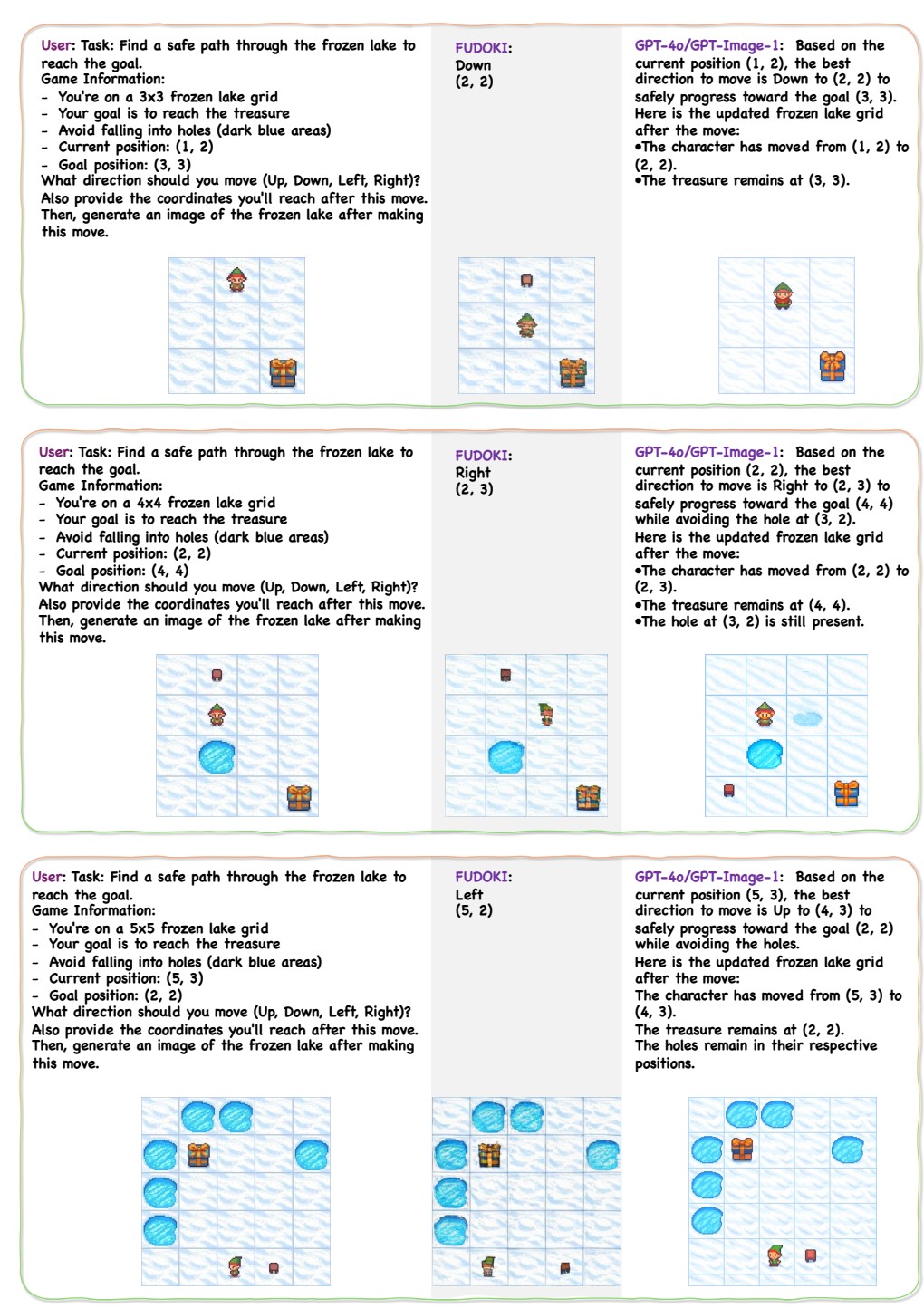

Figure 9: **Comparison of FUDOKI and GPT-4o/GPT-Image-1 on frozen lake maze navigation tasks**. GPT-4o/GPT-Image-1 offered well-reasoned textual outputs with safety and goal awareness but generated inconsistent visuals, even altering the maze (e.g., the third row). FUDOKI, by contrast, consistently produced valid directions and coherent visual updates aligned with task constraints, demonstrating stronger spatial consistency.

- **OCR** (428.0K, 11.5%): LLaVAR (59.3K) [61], SROIE (17.1K) [147], FUNSD (6.8K) [148], OCRVQA (80K) [149], MLHME-38K (30K) [150], Rendered Text (10.0K) [62], IIIT5K

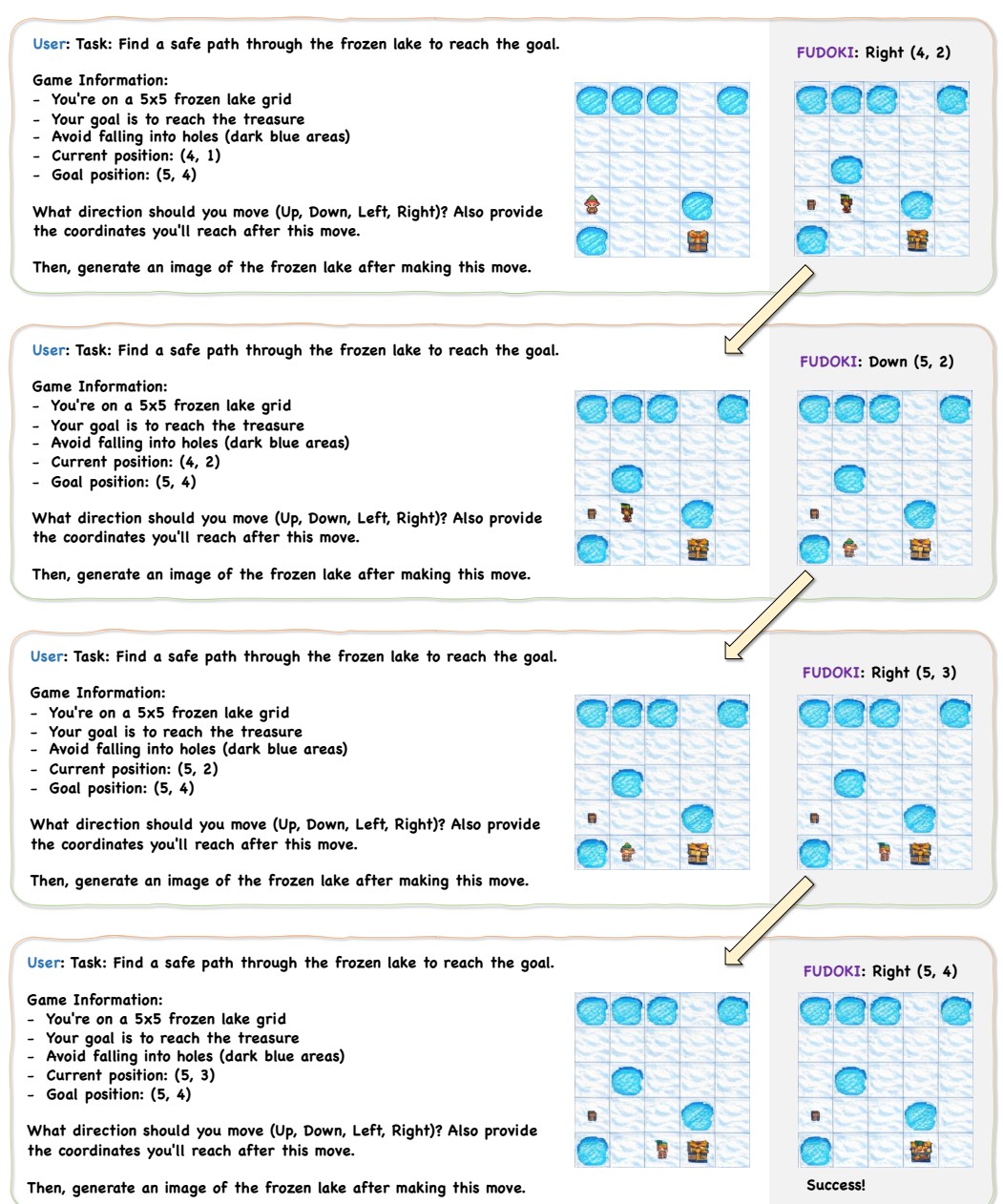

Figure 10: **FUDOKI successfully completed the full maze navigation task step by step.** Starting from the initial position at (4, 1), it sequentially selected safe moves—Right → Down → Right → Right—while avoiding holes and progressing toward the treasure at (5, 4). At each step, FUDOKI generated an updated image of the frozen lake, reflecting the character's new position and preserving the environment's structure, culminating in a successful arrival at the goal. Notably, in rows 2 through 4, the input images were taken directly from FUDOKI's previous outputs, demonstrating the model's ability to maintain coherent state tracking and visual continuity throughout the multistep decision-making process.

(6.0K) [151], HME100K (74.5K) [152], SynthDoG-EN (29.8K) [153], POIE (9.4K) [154], IAM (5.7K) [155], TextCaps (60.5K) [156], COCO-Text V2.0 (28.1K) [157], ChromeWriting (8.8K) [62], ORAND-CAR (2K) [158]

- **Document** (155.8K, 4.2%): DocVQA (122.4K) [63], FUNSD (6.8K) [148], Deepform (9.2K) [159], Kleister CharityAI (15.2K) [160], TAT-DQA (2.2K) [161]

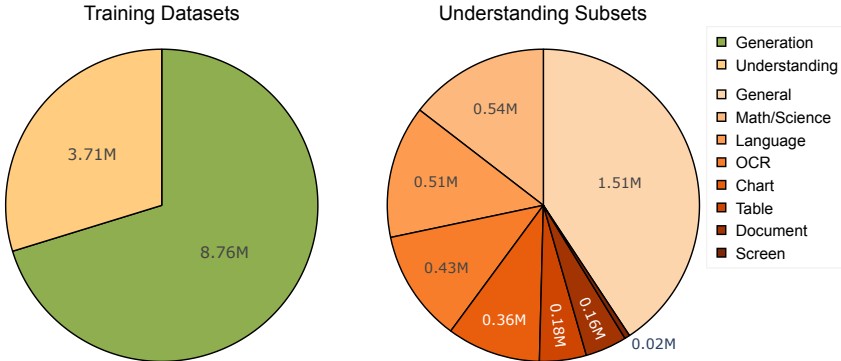

Figure 11: **Training Dataset Distribution.** The overall training data consists of 8.76M Generation samples (69%) and 3.86M Understanding samples (31%), as shown on the left. The right chart depicts the composition of the Understanding subset by category.

- **Table** (180.2K, 4.9%): TabFact (65.6K) [161], WikiTable (29.5K) [162], TabMWP (38.4K) [163], RoBUT WTQ (38.2K) [164], RoBUT SQA (8.5K) [164]
- **Chart** (362.6K, 9.8%): ChartQA (62.9K) [165], Chart2Text (27.0K) [64], PlotQA (10K) [166], DVQA (200K) [167], Infographic VQA (47.6K) [168], VisText (10.0K) [169], Diagram Image2Text (0.3K) [170], LRV Chart (1.8K) [171]
- **Screen** (24.6K, 0.7%): WebSRC (5.1K) [172], VisualMRC (19.5K) [65]
- **Math/Science** (544.9K, 14.7%): MAVIS (187.3K) [173], G-LLaVA (162.4K) [66], GeoQA+ (72.3K) [67], GeoMVerse (9.3K) [174], Geometry3K (3.0K) [175], MathVision (3.0K) [176], Cambrian Data Engine (50.8K) [177], Textbook QA (21.8K) [178], ScienceQA (19.2K) [179], AI2d (18.8K) [180]
- **Language** (510.2K, 13.7%): MathInstruct (81.5K) [181], Evol-Instruct (142.8K) [182], MathPlus (95.2K) [183], Magpie Pro (L3 MT) (50.0K) [68], ShareGPT4 (40.7K) [184], Magpie Pro (L3 ST) (50.0K) [68], Magpie Pro (Qwen2 ST) (50.0K) [68]

# E  Mathematical Formulations of Kinetic Optimal Velocity

To facilitate understanding, we use a simplified notation here and let $\mathcal{T}$ denote the finite discrete state space, with elements $x, z \in \mathcal{T}$ (in the main paper, we have $x^i, z^i \in \mathcal{T}$). A probability path is a time-varying distribution $p_t(x)$, and a velocity field $u_t(x, z)$ describes mass transport between states over time. In this way, we have the *Continuity Equation* as follows.

$$\dot{p}_t(x) + \mathrm{div}_x(j_t) = 0, \quad \forall x \in T$$

with the discrete divergence given by $\mathrm{div}_x(j_t) = \sum_{z \neq x} j_t(z, x) - \sum_{z \neq x} j_t(x, z)$ and $j_t(x, z)$ is the flux, defined by $j_t(x, z) = u_t(x, z)\, p_t(z)$, which represents the flow of probability mass from $z$ to $x$. In this way, the velocity can be obtained by $u_t(x, z) = \begin{cases} \frac{j_t(x,z)}{p_t(z)} & \text{if } p_t(z) > 0 \\ 0 & otherwise \end{cases}$ when $x \neq z$ and $u_t(z, z) = -\sum_{x \neq z} u_t(x, z)$ to ensure the rate condition in Eq. 2. With such notations, we expect to minimize the kinetic energy during the flow process, namely,

$$\min_{p_t, j_t} \int_0^1 \sum_{x \neq z} w_t(x, z) \frac{j_t(x, z)^2}{p_t(z)}\, dt$$

subject to:

- Continuity Equation: $\mathrm{div}_x(j_t) = -\dot{p}_t(x)$
- Non-negativity of the flux: $j_t(x, z) \geq 0 \quad \forall x \neq z$
- Boundary conditions: $p_0 = p, \quad p_1 = q$

Here, $w_t(x, z) > 0$ is a problem-specific weight controlling the "cost" of mass moving from $z$ to $x$. As evidenced in [38], when $p_t$ is given and let $w_t(x, z) = 1/p_t(x)$, the kinetic optimal solution can be obtained via $j_t^\star(x, z) = [p_t(z)\dot{p}_t(x) - \dot{p}_t(z)p_t(x)]_+ \quad \forall x \neq z$. In this way, if we apply this kinetic optimal $j_t^\star(x, z)$ for the probability path in Eq. 4, we can obtain the velocity defined in Eq. 5.

## F    Limitations and Broader Impacts

**Limitations**. Despite its promising results, FUDOKI also presents several limitations that warrant further investigation. First, despite the advantages of discrete flow matching—such as being agnostic to token order and compatible with bidirectional Transformers—the current implementation requires the sequence length to be fixed prior to sampling. This constraint limits flexibility in generation and makes dynamic-length outputs challenging. A promising direction for future work is to extend the sampling scheme to support variable-length generation, which would broaden the applicability of the model across open-ended tasks and enhance the flexibility on the computational cost during inference. Besides, as shown in Fig. 12, while FUDOKI shows strong performance, it still faces challenges under certain scenarios, such as performing text-to-image generation given complex prompts or prompts involving rendering specific texts in images, as well as performing visual understanding tasks that demand expert-level reasoning and domain-specific knowledge.

**Image Understanding**            **Image Generation**

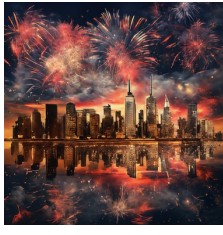
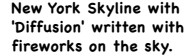
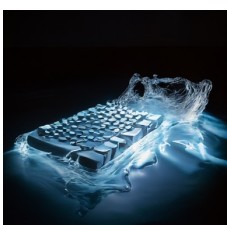

New York Skyline with 'Diffusion' written with fireworks on the sky.

A keyboard made of water, the water is made of light, the light is turned off.

Figure 12: **Examples of failed cases on visual understanding and generation.** While FUDOKI demonstrated strong performance, it still struggled with harder tasks—such as generating images from complex prompts involving specific texts, and understanding visuals that require expert-level knowledge.

**Broader Impacts**. FUDOKI introduces a novel paradigm for unified multimodal modeling that departs from the long-dominant autoregressive approach, potentially redefining how future multimodal systems are designed. By leveraging discrete flow matching with metric-induced probability paths, FUDOKI enables controllable and interpretable generation processes, which could prove valuable in critical applications such as education, embodied AI, and autonomous driving. Its iterative, self-correcting refinement process aligns well with human reasoning patterns and may support safer, more reliable AI agents in domains requiring high precision, such as medicine and law. Furthermore, FUDOKI's unified architecture for both understanding and generation fosters more integrated, general-purpose agents—an important step toward realizing practical artificial general intelligence (AGI). However, as with any generative technology, ethical considerations around bias, misuse, and content safety must be carefully addressed as adoption scales.

