# OpenReview forum: "FUDOKI: Discrete Flow-based Unified Understanding and Generation via Kinetic-Optimal Velocities"
_NeurIPS.cc/2025/Conference — NeurIPS 2025 spotlight_

### Official Review · Reviewer_vmHC · 2025-06-27

**Clarity:** 3
**Significance:** 3
**Originality:** 3
**Rating:** 4
**Confidence:** 4

**Summary:**

The paper introduces FUDOKI, the first unified vision-language model that replaces the standard autoregressive (AR) decoder with Discrete Flow Matching (DFM) decoding. Tokens are noised along a metric-induced probability path in embedding space, and a kinetic-optimal velocity field is learned so the model iteratively transports the corrupted sequence toward the target, allowing bidirectional attention and step-wise self-correction. Trained on Jauns-1B, FUDOKI achieves performance comparable to SOTA AR-based MLLMs across both visual understanding and image generation tasks,

**Questions:**

The questions are the same as the weaknesses we discussed above. My main concern is whether the motivation for introducing DFM is sufficiently justified. While the self-correction capability is certainly appealing, the paper lacks quantitative experiments to demonstrate the effectiveness of this capability within the unified model. Addressing this issue would  improve my rating of the paper.

**Ethical Concerns:**

["NO or VERY MINOR ethics concerns only"]

**Final Justification:**

The author's new experiment, a quantitative comparison of the self-correction capabilities of AR and FUDOKI, resolved my biggest concern, so I have raised my score for this paper to 4.

**Limitations:**

yes

**Paper Formatting Concerns:**

There are no obvious formatting issues in the paper.

**Quality:**

3

**Strengths And Weaknesses:**

**Strengths**
1. This paper presents the first unified multimodal generation model based on Discrete Flow Matching, introducing DFM techniques into this domain.
2. The figures in this paper are clear and effectively illustrate the differences between the proposed method and existing approaches.

**Weaknesses**
1. The novelty of this paper is somewhat limited. The self-correction capability is an inherent property of DFM models, and while the paper applies it to a unified model, it does not demonstrate any unique advantages in this setting. In my view, this reflects a lack of sufficient motivation.
2. The results presented in this paper are also not particularly strong. Despite initializing from Janus-1B weights, the model undergoes an additional 43,000 GPU hours of training—roughly equivalent to training a 7B autoregressive MLLM from scratch. However, the final performance only matches that of existing AR-based models, and on many benchmarks, the results are not especially competitive.
3. The main motivation of this paper lies in the self-correction capability of DFM. However, this is only demonstrated through a few qualitative visualizations. While these are intuitive, they lack statistical rigor by academic review standards. The paper does not provide quantitative experiments to convincingly show that self-correction is truly effective and important for MMU tasks.
4. The Related Work section of this paper lacks citations to some relevant papers, such as Muse-vl, LlamaFusion and OmniMamba. Including references to these papers would help make this section more comprehensive.

---

> ### Author Rebuttal · Authors · 2025-07-30
>
> Thank you for your great efforts on the review and constructive comments.
> We will try our best to answer all your questions.
>
> **Please let us know if you are not satisfied with our answers or have further questions, so that we can get back to you as soon as possible.**
>
> $\textcolor{blue}{\textsf{Q1: About motivations of FUDOKI.}}$
>
>
> > "The novelty of this paper is somewhat limited. The self-correction capability is an inherent property of DFM models, and while the paper applies it to a unified model, it does not demonstrate any unique advantages in this setting. In my view, this reflects a lack of sufficient motivation."
>
> A1: Thank you for your comments. While self-correction is indeed an inherent property of DFM models, we would like to clarify that the motivation behind FUDOKI extends beyond this single aspect. As stated in our introduction, our primary motivation is to **challenge the dominance of the prevailing autoregressive (AR) paradigm** in unified multimodal modeling.
> To this end, FUDOKI is the first general-purpose unified multimodal model built entirely on **discrete flow matching**, providing several unique advantages over AR approaches that could benefit multimodal tasks, such as **bidirectional information integration** and **parallel, iterative inference**.
> While **self-correction** is also one consequence of adopting discrete flow matching, our main motivation is to explore **a new, flexible and generalizable modeling paradigm** for unified multimodal models.
> We appreciate the reviewer’s suggestion and will further clarify these discussions in the revised manuscript.
>
>
> $\textcolor{blue}{\textsf{Q2: Ask for quantitative experiments for evaluating the self-correction capability of FUDOKI.}}$
>
>
> > "The main motivation of this paper lies in the self-correction capability of DFM. However, this is only demonstrated through a few qualitative visualizations. While these are intuitive, they lack statistical rigor by academic review standards. The paper does not provide quantitative experiments to convincingly show that self-correction is truly effective and important for MMU tasks." "My main concern is whether the motivation for introducing DFM is sufficiently justified. While the self-correction capability is certainly appealing, the paper lacks quantitative experiments to demonstrate the effectiveness of this capability within the unified model."
>
> A2: Thank you for your comments. We have followed your suggestions and conducted new experiments, where we quantitatively evaluated the self-correcting capabilities of our proposed FUDOKI and performed comparisons with the AR-based models.
> In experiments, both FUDOKI and AR-based models were tasked with correcting baseline responses where necessary. The baseline responses were obtained from Janus-Pro-1B on the MMVet benchmark, using the OpenCompass VLMEvalKit codebase [cite 1]. To assess their correction abilities,
>
> - For AR-based models, we appended the following prompt to the original prompt: "Your original response is: \<placeholder\>. Please correct it if needed. Otherwise, you may keep it the same." The models were then evaluated on their ability to revise or retain the response as appropriate.
> - For FUDOKI, we initialized the responses with the baseline responses (rather than uniformly-sampled noise tokens) and performed iterative refinements over 32 steps, as described in the paper.
>
> As shown in Table A, FUDOKI achieved the highest performance improvement, while Janus-Pro-1B’s performance declined and Janus-Pro-7B showed less increase, despite its larger model size than ours.
> We attribute such results to the increased context length introduced by the baseline responses, which may distract the AR-based model’s focus.
> This further highlights the limitations of the AR paradigm for effective self-correction.
>
> **Table A: Quantitative comparisons between the AR-based models and our proposed FUDOKI in terms of the self-correcting capabilities.**
>
> | Model | Baseline |+Janus-Pro-1B to correct |+Janus-Pro-7B to correct | +FUDOKI to correct |
> |:-----|:-------:|:------:|:-----:|:-----:|
> |  MMVet  |  37.98 | 36.33 (-1.65) | 38.30 (+0.32)|**38.53 (+0.55)**|
>
>
> [cite 1] Duan, Haodong et al. Vlmevalkit: An open-source toolkit for evaluating large multi-modality models. In ACMMM 2024.
>
>
>
> $\textcolor{blue}{\textsf{Q3: About the results and training costs.}}$
>
>
> > "The results presented in this paper are also not particularly strong. Despite initializing from Janus-1B weights, the model undergoes an additional 43,000 GPU hours of training—roughly equivalent to training a 7B autoregressive MLLM from scratch. However, the final performance only matches that of existing AR-based models, and on many benchmarks, the results are not especially competitive."
>
> A3: Thank you for your comments. Due to computational constraints, all experiments in our paper were actually conducted on NVIDIA V100 GPUs.
> As such, training large-scale 7B autoregressive MLLM baselines was not feasible within our resources.
> However, we would like to emphasize that our primary goal was to demonstrate the feasibility and potential of discrete flow matching (DFM) in multimodal modeling, rather than to surpass all existing AR-based models. While our current results are on par with AR-based models, we believe this is promising given our smaller model size and limited compute budget.
> We expect that, with larger models, more extensive training, and improved scaling strategies—such as those used in recent AR-based models like BAGEL [cite 1]—the advantages of DFM-based unified models, including inherent bidirectional reasoning and self-correction, will become more evident.
> We appreciate the reviewer’s feedback and will clarify these points in the revised manuscript. In future work, we aim to further explore scaling up DFM models if computational resources permit.
>
> [cite 1] Deng C, et al. Emerging properties in unified multimodal pertaining. Preprint, 2025.
>
>
> $\textcolor{blue}{\textsf{Q4: Relevant studies are missing.}}$
>
>
> > "The Related Work section of this paper lacks citations to some relevant papers, such as Muse-vl, LlamaFusion and OmniMamba. Including references to these papers would help make this section more comprehensive."
>
> A4: Thank you for your feedback. We will cite and discuss all these works to make the Related Work section more comprehensive in our revised paper.

---

> > ### Comment · Reviewer_vmHC · 2025-08-01
> > **Reply to the rebuttal**
> >
> > Thank you for your reply. It has resolved my concern. I will increase my rating to 4.

---

> > > ### Author Response · Authors · 2025-08-01
> > > **Thanks for your feedback**
> > >
> > > Dear Reviewer vmHC,
> > >
> > > Thank you for your feedback! We greatly appreciate your constructive suggestions for improving our work and will revise our manuscript accordingly.
> > >
> > > Thank you again for your dedicated time and effort in reviewing our paper.
> > >
> > > Best regards,
> > >
> > > Authors

---

### Official Review · Reviewer_sZnw · 2025-06-28

**Clarity:** 4
**Significance:** 4
**Originality:** 4
**Rating:** 5
**Confidence:** 4

**Summary:**

This paper presents FUDOKI, the first unified multimodal generative and understanding model based entirely on discrete flow matching (DFM). The model supports both image and text modalities, featuring a mask-free, bidirectional, and dynamically revisable generation process. Experiments demonstrate that FUDOKI matches or surpasses state-of-the-art autoregressive MLLMs in visual understanding and image generation benchmarks.

**Questions:**

The paper does not compare FUDOKI with recent strong unified models like Bagel [1]; could the authors briefly discuss FUDOKI's relative strengths or weaknesses compared to it?
[1] Unified Model for Multimodal Understanding and Generation

**Ethical Concerns:**

["NO or VERY MINOR ethics concerns only"]

**Final Justification:**

Authors have resolved most of my concerns. I will remain my rating.

**Limitations:**

The paper primarily evaluates standard single-image tasks. Broader validation on diverse, practical, or multi-image scenarios, and deeper discussion of societal impacts, would strengthen the work.

**Quality:**

4

**Strengths And Weaknesses:**

**Strengths**
- Technically sound and well-motivated methodology.
- Unified framework for multimodal generation and understanding.
- Supports dynamic token revision during generation, improving flexibility.
- Strong empirical results on standard benchmarks.
- Novel application of discrete flow matching with kinetic-optimal velocity.

**Weaknesses**
- The paper does not provide experimental results or analysis for more complex multimodal scenarios, such as multi-image conditioning or compositional tasks.
- No comparison is made with recent strong unified AR+DiT models like Bagel, leaving it unclear how FUDOKI performs against state-of-the-art baselines on unified generation and understanding.

---

> ### Author Rebuttal · Authors · 2025-07-30
>
> Thank you for your great efforts on the review and constructive comments.
> We will try our best to answer all your questions.
>
> **Please let us know if you are not satisfied with our answers or have further questions, so that we can get back to you as soon as possible.**
>
> $\textcolor{blue}{\textsf{Q1: Ask about analysis of FUDOKI on more complex multimodal scenarios and its societal impacts.}}$
>
>
> > "The paper does not provide experimental results or analysis for more complex multimodal scenarios, such as multi-image conditioning or compositional tasks." "The paper primarily evaluates standard single-image tasks. Broader validation on diverse, practical, or multi-image scenarios, and deeper discussion of societal impacts, would strengthen the work."
>
> A1: Thank you for your comments.
> As an initial step, we conducted preliminary experiments on interleaved image-text generation, *i.e.*, Maze Navigation, which involves one input image for understanding and one output image for generation. The results are presented in Figures 9–10 of the supplementary materials.
> However, due to computational constraints—specifically, all experiments in this work were performed on NVIDIA V100 GPUs—we were unable to extend training to more complex multimodal scenarios involving multiple images at a large scale. We recognize the importance of broader validation on such tasks and plan to explore these directions in future work if more computational resources are available.
>
> Regarding societal impacts, we agree that a deeper discussion is important.
> As multimodal models become more capable, their potential applications—ranging from assistive technology to content creation—grow, but they also bring risks of misuse, bias amplification, and challenges in content authenticity verification. While our work focuses on advancing technical foundations, we recognize the importance of continually assessing these broader impacts and will include such discussions in our revised paper.
>
>
> $\textcolor{blue}{\textsf{Q2: Ask for comparisons with recent strong unified models, such as Bagel [cite 1].}}$
>
>
> > "No comparison is made with recent strong unified AR+DiT models like Bagel, leaving it unclear how FUDOKI performs against state-of-the-art baselines on unified generation and understanding." "The paper does not compare FUDOKI with recent strong unified models like Bagel [1]; could the authors briefly discuss FUDOKI's relative strengths or weaknesses compared to it? [1] Unified Model for Multimodal Understanding and Generation"
>
> A2: Thank you for your comments.
> Bagel [cite 1] is a very strong recent advance in unified multimodal understanding and generation.
> While both FUDOKI and Bagel aim for unified multimodal modeling, they are based on fundamentally different generative paradigms and architectural choices.
> Specifically, Bagel employs a large Mixture-of-Transformer-Experts (MoT) architecture and follows the **autoregressive (AR) modeling** paradigm, enabling it to efficiently scale with massive, carefully structured interleaved multimodal data.
> In contrast, FUDOKI is the first general-purpose unified multimodal model built entirely on **discrete flow matching**, which allows for bidirectional information integration and iterative self-correction during generation.
> In terms of empirical performance, Bagel demonstrates strong results on both multimodal generation and understanding, including advanced tasks such as free-form image manipulation.
> We acknowledge that FUDOKI currently lags behind Bagel, which can be attributed mainly to Bagel’s novel data scaling strategies and substantially larger model size (14B parameters for Bagel *vs.* 1.5B for FUDOKI).
> We appreciate the reviewer bringing this to our attention and will include a more detailed discussion and citation of Bagel in our revised paper.
> We will also explore integrating similar scaling approaches in future work if more computational resources are available.
>
>
> [cite 1] Deng C, et al. Emerging properties in unified multimodal pertaining. Preprint, 2025.

---

> ### Comment · Reviewer_sZnw · 2025-08-04
> **Reply to the rebuttal**
>
> Thank you for your reply. It has resolved most of my concerns. I will remain my rating.

---

### Official Review · Reviewer_Zj7H · 2025-07-01

**Clarity:** 4
**Significance:** 3
**Originality:** 3
**Rating:** 4
**Confidence:** 3

**Summary:**

The paper introduces FUDOKI, a unified multimodal model for both visual understanding and image generation, built on a discrete flow matching paradigm rather than traditional autoregressive (AR) methods. This shift addresses key AR limitations, such as rigid generation order and weaker reasoning or self-correction.

FUDOKI refines outputs via metric-based probability paths and kinetically optimal velocity fields, allowing for richer bidirectional context and improved generation quality. To ease training, it starts from a pre-trained AR MLLM and transitions gradually to the flow-based framework.

Experiments show that FUDOKI matches or exceeds state-of-the-art AR models on tasks like visual question answering and text-to-image generation. Inference-time strategies (e.g., sampling multiple outputs) further enhance performance, highlighting the model’s flexibility and potential for continued improvement.

**Questions:**

1.Could the authors clarify the inference speed and step count required by FUDOKI versus an AR model? For instance, how many refinement iterations are typically needed for generation.

2.Do the authors think *any* strong AR language model could be converted to a flow-based model via their adaptation strategy? It would be insightful to discuss if the success depends on specific properties of the chosen base model (Janus-1.5B), or if one could take a larger model or different architecture and apply the same paradigm.

3.The paper introduces multiple novel components. It would strengthen the work to see more ablations.

**Ethical Concerns:**

["NO or VERY MINOR ethics concerns only"]

**Final Justification:**

The rebuttal has addressed most of my concerns, and I would therefore keep my positive rating.

**Limitations:**

Yes.

**Paper Formatting Concerns:**

No.

**Quality:**

3

**Strengths And Weaknesses:**

Strengths:

1.It is the first work to build a general-purpose vision–language model entirely on discrete flow matching, which advance beyond prior diffusion models relying on basic mask-based corruption. It addresses key limitations of autoregressive multimodal models.

2.The paper is clearly written, with a well-structured presentation. Section 2 provides a solid overview of discrete flow matching, while Section 3 clearly explains the proposed extensions—metric-induced paths and kinetically optimal velocity fields. The distinctions from prior methods, such as mask-based diffusion versus their continuous refinement approach, are effectively illustrated.

3.In terms of experiments, the paper provides thorough validation. FUDOKI is evaluated on a wide range of well-known tasks and benchmarks.



Weaknesses:

1.Inference speed remains unclear. While the model benefits from iterative refinement and self-correction and show quality-speed Trade-off, it’s not obvious how this translates to actual inference efficiency. There is no direct comparison with optimized AR models in terms of speed or number of steps required to reach similar output quality.

2.Lack of ablations. While the paper introduces several novel components—such as metric-induced probability paths, kinetically optimal velocity fields, and test-time inference scaling—there is no systematic ablation to isolate their individual impact. For example, it would be informative to see how performance changes if a simpler mask-based path replaces the metric-induced one.

3.Limited Evaluation Scope. While FUDOKI shows promising generalization ability and robustness across several visual generation and understanding tasks, its effectiveness on more challenging reasoning tasks or open-ended question answering scenarios remains underexplored. Further evaluation is needed to confirm whether its strengths extend to these more complex settings.

---

> ### Author Rebuttal · Authors · 2025-07-30
>
> Thank you for your great efforts on the review and constructive comments.
> We will try our best to answer all your questions.
>
> **Please let us know if you are not satisfied with our answers or have further questions, so that we can get back to you as soon as possible.**
>
> $\textcolor{blue}{\textsf{Q1: Ask about the inference speed comparison with the AR model.}}$
>
> > "Inference speed remains unclear. While the model benefits from iterative refinement and self-correction and show quality-speed Trade-off, it’s not obvious how this translates to actual inference efficiency. There is no direct comparison with optimized AR models in terms of speed or number of steps required to reach similar output quality." "Could the authors clarify the inference speed and step count required by FUDOKI versus an AR model? For instance, how many refinement iterations are typically needed for generation."
>
> A1: Thank you for your comments. Please refer to Figure 5 (right) in the main paper, which also compares the inference performance of FUDOKI with the autoregressive (AR) baseline, Janus-Pro-1B (with KV cache enabled).
> All experiments, including this comparison, were conducted on the NVIDIA V100 GPU in this paper.
> In the figure, the blue dashed line (to the right vertical axis) indicates the GenEval score of Janus-Pro-1B, while the red dashed line (to the left vertical axis) shows its inference speed, measured in the number of generated images per minute.
> We kindly direct the reviewer’s attention to the intersection point of the green arrows in Figure 5 (right). This intersection marks the point where FUDOKI achieves a significant speed advantage over the AR baseline (as the red solid line exceeds the red dashed line) and comparable output quality (where the blue solid line meets the blue dashed line).
> Moreover, in practice, we find that FUDOKI typically requires 10 steps to generate plausible images, whereas the AR model, Janus-Pro-1B, requires 576 steps. This demonstrates the efficiency of our approach in terms of both inference speed and step count.
>
> $\textcolor{blue}{\textsf{Q2: Ask about more ablation studies of FUDOKI.}}$
>
> > "Lack of ablations. While the paper introduces several novel components—such as metric-induced probability paths, kinetically optimal velocity fields, and test-time inference scaling—there is no systematic ablation to isolate their individual impact. For example, it would be informative to see how performance changes if a simpler mask-based path replaces the metric-induced one." "The paper introduces multiple novel components. It would strengthen the work to see more ablations."
>
> A2: Thank you for your comments.
> Due to time and computational resource constraints, we were unable to complete the training of a mask-based FUDOKI variant during the rebuttal phase.
> Nevertheless, the effectiveness of metric-induced probability paths has been systematically evaluated in its original work (*e.g.*, Table 3 in [cite 1]).
> Additionally, FUDOKI achieves performance comparable to recent mask-based discrete diffusion MLLMs, such as MMaDA [cite 2], even at a smaller scale (8B parameters for MMaDA *vs.* 1.5B for FUDOKI), highlighting the competitiveness of our approach.
> We appreciate the importance of detailed ablation studies and will include such analyses in future work.
>
> [cite 1] Shaul, Neta, et al. Flow matching with general discrete paths: A kinetic-optimal perspective. In ICLR 2025.
>
> [cite 2] Yang, Ling, et al. Mmada: Multimodal large diffusion language models. Preprint, 2025.
>
> $\textcolor{blue}{\textsf{Q3: Ask for evaluations of FUDOKI on more challenging reasoning tasks.}}$
>
> > "Limited Evaluation Scope. While FUDOKI shows promising generalization ability and robustness across several visual generation and understanding tasks, its effectiveness on more challenging reasoning tasks or open-ended question answering scenarios remains underexplored. Further evaluation is needed to confirm whether its strengths extend to these more complex settings."
>
> A3: Thank you for your comments. We have followed your suggestions and conducted new experiments, where we evaluated our proposed FUDOKI on a more challenging mathematical reasoning benchmark, MathVista (testmini) [cite 1].
> As shown in Table A, we find that FUDOKI achieved the best performance compared to AR-based models at the same scale.
> We attribute this improvement to FUDOKI's discrete flow matching framework, which leverages bidirectional context modeling to facilitate complex reasoning.
>
> **Table A: Performance comparisons on the MathVista benchmark.**
>
> | Model | Janus-1.5B | Janus-Pro-1B| FUDOKI |
> |:-----|:-------:|:------:|:-----:|
> |  MathVista  |  32.4 | 35.1 | **38.6** |
>
> [cite 1] Lu, Pan, et al. Mathvista: Evaluating mathematical reasoning of foundation models in visual contexts." In ICLR 2024.
>
> $\textcolor{blue}{\textsf{Q4: Discussions on the base model of FUDOKI.}}$
>
> > "Do the authors think any strong AR language model could be converted to a flow-based model via their adaptation strategy? It would be insightful to discuss if the success depends on specific properties of the chosen base model (Janus-1.5B), or if one could take a larger model or different architecture and apply the same paradigm."
>
> A4: Thank you for your comments. We have followed your suggestion and conducted new experiments, using another AR model, Qwen2-VL-2B-Instruct, as the base model.
> However, we observed that it converged significantly more slowly on the image generation task compared to Janus-1.5B.
> This observation suggests that the original AR model’s integrated capabilities for both understanding and generation are important for the success of the adaptation strategy.

---

### Official Review · Reviewer_WC1Q · 2025-07-01

**Clarity:** 3
**Significance:** 2
**Originality:** 2
**Rating:** 4
**Confidence:** 4

**Summary:**

The paper introduce a multimodal model based on kinetic flow matching. The author leverage pretrained AR model and the formulation from [1] to generate both text and image. For efficient finetuning from AR models, they also adopt strategies from [2] i.e., shifting, and discard the time embedding layers. The paper demonstrate the performance of their models on image understanding and generation are competitive with AR-based approaches.





[1] Flow Matching with General Discrete Paths: A Kinetic-Optimal Perspective (ICLR 2025)

[2] Scaling Diffusion Language Models via Adaptation from Autoregressive Models (ICLR 2025)

**Questions:**

- Including a GIF that visualizes the text‐generation process would add valuable insight. From my experience, absorbing‐diffusion models like LLaDA and Dream often behave largely autoregressively, so a clear demonstration of their bidirectional generation capability would be especially informative.
- It would be helpful if the author add the performance of model with different number of denoising steps.
- Are experiments run on H100s? Why don't the author fine-tune existing (larger) language diffusion baselines such as LLaDA or Dream (as I believe 43k gpu hours is quite substantial and enough for fine-tuning 7B models)? how straightforward it is to adapt an absorbing diffusion model to the kinetic diffusion?

**Ethical Concerns:**

["NO or VERY MINOR ethics concerns only"]

**Final Justification:**

After reading the other review and rebuttal I still believe the novelty is a little limited but the execution is good to me. Therefore, I keep my initial score and lean toward acceptance.

**Limitations:**

Yes

**Quality:**

3

**Strengths And Weaknesses:**

### Strengths:
- The model performs competitive with AR models of similar scale on various benchmarks.
- The paper is well-written and the experiment is well-designed

### Weaknesses:
- The algorithm itself is not novel:  the kinetic flow matching formulation is introduced in [1], the idea of adopting pre-trained AR for discrete diffusion is proposed in [2] (the author use technique from that paper as well), discarding time embedding layers is proposed and investigated in [3] and previous absorbing diffusion. The novelty mostly lie in the fact that the author extend that to visual domain. In my point of view, this makes their contribution borderline.
- Similar to previous language diffusion models, the proposed approach operates under a fixed compute budget, which is limited for multimodal generation. For instance, a $1024\times 1024$ pixel image translates to roughly $4096$ tokens, allocate a "reasonable" number of generated tokens during inference phase is not straightforward e.g., $4096$ tokens are generally not efficient for language generation task.
- The model use two type of image encoders (SigLIP and VAE), thus, training pipeline could be rather cumbersome. For instance, multi-turn chat with multiple images, which is not straightforward to separate the understanding and generation task. In addition, it would be difficult to extend image generation tasks beyond text-to-image e.g., (multiple) image to image generation.


[3] Is Noise Conditioning Necessary for Denoising Generative Models? (ICML 2025)

---

> ### Author Rebuttal · Authors · 2025-07-30
>
> Thanks for your great efforts on the review and constructive comments.
> We will try our best to answer all your questions.
>
> **Please let us know if you are not satisfied with our answers or have further questions, so that we can get back to you as soon as possible.**
>
> $\textcolor{blue}{\textsf{Q1: Ask about the contributions of FUDOKI.}}$
>
> > "The algorithm itself is not novel: the kinetic flow matching formulation is introduced in [1], the idea of adopting pre-trained AR for discrete diffusion is proposed in [2] (the author use technique from that paper as well), discarding time embedding layers is proposed and investigated in [3] and previous absorbing diffusion. The novelty mostly lie in the fact that the author extend that to visual domain. In my point of view, this makes their contribution borderline."
>
> A1: Thank you for your comments and for highlighting the relevant prior work.
> We acknowledge that our approach builds upon several important previous studies. However, we would like to emphasize the key contributions of our work as follows:
>
> - To the best of our knowledge, FUDOKI is **the first unified MLLM built entirely on discrete flow matching for both visual understanding and generation tasks**. This approach brings several unique advantages to multimodal modeling, including **bidirectional information integration**, **parallel inference**, and **iterative self-correction through refinements**. In addition, unlike recent approaches such as MMaDA [cite 1] and LaViDa [cite 2], FUDOKI does not rely on mask tokens.
> - Our experiments demonstrate that **FUDOKI achieves performance on par with autoregressive models at the same scale**. Additionally, we also provide the first empirical validation of inference scaling within the discrete flow matching framework.
> - **We will release both our model and code to benefit and advance the open-source community**, especially considering that the open-source implementations of discrete flow matching are still limited.
>
> We hope these points clarify the contributions and significance of our work, and we believe that our findings and open-sourced resources can inspire and facilitate further research in this area.
>
> [cite 1] Yang, Ling, et al. Mmada: Multimodal large diffusion language models. Preprint, 2025.
>
> [cite 2] Li, Shufan, et al. Lavida: A large diffusion language model for multimodal understanding. Preprint, 2025.
>
>
> $\textcolor{blue}{\textsf{Q2: Ask about the fixed compute budget of the discrete flow matching (DFM) framework.}}$
>
> > "Similar to previous language diffusion models, the proposed approach operates under a fixed compute budget, which is limited for multimodal generation. For instance, a 1024×1024 pixel image translates to roughly 4096 tokens, allocate a "reasonable" number of generated tokens during inference phase is not straightforward e.g., 4096 tokens are generally not efficient for language generation task."
>
> A2: Thank you for your careful review. We acknowledge that the standard DFM framework requires specifying a fixed compute budget during inference.
> However, recent work—such as EditFlow [cite 1] and DreamOn [cite 2]—has begun to address this issue by introducing edit operations that enable more flexible and dynamic adjustment of output lengths in non-autoregressive frameworks. Additionally, advances in image representation, such as using fewer tokens for images [cite 3–4], offer promising directions for improving generation efficiency.
> We consider incorporating these techniques into FUDOKI a promising avenue for future research, which could further enhance its flexibility and applicability to diverse multimodal tasks.
>
> [cite 1] Havasi, Marton, et al. Edit Flows: Flow Matching with Edit Operations. Preprint, 2025.
>
> [cite 2] Wu, Zirui, et al. DreamOn: Diffusion Language Models For Code Infilling Beyond Fixed-size Canvas. Blog, 2025.
>
> [cite 3] Yu, Qihang, et al. An image is worth 32 tokens for reconstruction and generation." In NIPS 2024.
>
> [cite 4] Bachmann R, et al. FlexTok: Resampling Images into 1D Token Sequences of Flexible Length. In ICML 2025.
>
>
>
> $\textcolor{blue}{\textsf{Q3: Ask about the limitations of using separate encoders for images.}}$
>
> > "The model use two type of image encoders (SigLIP and VAE), thus, training pipeline could be rather cumbersome. For instance, multi-turn chat with multiple images, which is not straightforward to separate the understanding and generation task. In addition, it would be difficult to extend image generation tasks beyond text-to-image e.g., (multiple) image to image generation."
>
> A3: Thank you for your comments. We acknowledge that employing two types of image encoders adds complexity to the training pipeline. This design choice was inherited from Janus, aiming to leverage the strengths of each encoder in their respective domains.
> In contrast, recent studies [cite 1-2] have proposed to develop unified image representations for both understanding and generation, which could simplify the pipeline and enhance the model’s flexibility across various multimodal tasks, such as the mentioned multi-turn chat with multiple images and (multiple) image to image generation.
> We appreciate the reviewer’s observation and will explore this in future iterations of our system.
>
> [cite 1] Qu L, et al. Tokenflow: Unified image tokenizer for multimodal understanding and generation. In CVPR 2025.
>
> [cite 2] Ma C, et al. Unitok: A unified tokenizer for visual generation and understanding. Preprint, 2025.
>
>
>
> $\textcolor{blue}{\textsf{Q4: Ask about the GIF visualizations of the text generation process.}}$
>
> > "Including a GIF that visualizes the text‐generation process would add valuable insight. From my experience, absorbing‐diffusion models like LLaDA and Dream often behave largely autoregressively, so a clear demonstration of their bidirectional generation capability would be especially informative."
>
> A4: Thank you for your comments.
> During the submission phase, we included GIF visualizations of the text generation process on our project page, as referenced in the footnote on Page 1 of the paper.
> In accordance with the NeurIPS 2025 rebuttal policy, the project page has not been updated during the rebuttal phase.
> Additional examples can also be found in Figure 8 of the supplementary materials. Notably, these visualizations demonstrate that our generation process exhibits less autoregressive behavior, showcasing the model’s bidirectional generation capability.
>
> $\textcolor{blue}{\textsf{Q5: Ask about the performance of FUDOKI given different inference steps.}}$
>
> > "It would be helpful if the author add the performance of model with different number of denoising steps."
>
> A5: Thank you for your comments. Please see Figure 5 (right) in the main paper, showcasing the model's performance on the GenEval benchmark with different inference steps (*i.e.*, the blue line).
> Results show that with more timesteps, FUDOKI achieved better generation performance.
>
> $\textcolor{blue}{\textsf{Q6:}}$
>
> > "Are experiments run on H100s? Why don't the author fine-tune existing (larger) language diffusion baselines such as LLaDA or Dream (as I believe 43k gpu hours is quite substantial and enough for fine-tuning 7B models)? how straightforward it is to adapt an absorbing diffusion model to the kinetic diffusion?."
>
> A6: Thank you for your comments. Due to our computational
> constraints, all experiments in the paper were actually conducted on NVIDIA V100 GPUs. As such, fine-tuning larger language diffusion baselines was not feasible within our resources. Regarding model adaptation, in principle, converting an absorbing diffusion model to the kinetic discrete flow matching framework does not require specific architectural modifications. We consider this a promising direction and will explore this in future work if more computational resources are available.

---

> > ### Comment · Reviewer_WC1Q · 2025-08-07
> > **Reviewer's response**
> >
> > Thanks for the detailed rebuttal, most of my concerns are addressed.

---

### Official Review · Reviewer_CdAa · 2025-07-02

**Clarity:** 3
**Significance:** 3
**Originality:** 2
**Rating:** 4
**Confidence:** 4

**Summary:**

This paper introduces a unified model based on discrete flow matching for both generation and understanding tasks by leveraging metric-induced probability paths with kinetic optimal velocities. The model achieves promising results across multiple benchmark datasets.

**Questions:**

Questions:

Although the paper applies flow matching to unified models, it uses the AR-based Janus as the base model for training. Could the authors explain why they did not choose MaskGIT or another diffusion model instead?

Beyond benchmark tests, it would strengthen the paper to include other applications. Could the authors provide results on interleaved image generation or on image editing tasks?

**Ethical Concerns:**

["NO or VERY MINOR ethics concerns only"]

**Final Justification:**

I do not have more concerns. I think this paper propose a effective architecture to unify two tasks using discrete diffusion. However, I think the novelty is a little limited and the motivation using discrete diffusion is a little uncertain. Finally, I stay positive attitude torwards this work and I give boderline accept.

**Limitations:**

It can be seen in weeknesses.

**Paper Formatting Concerns:**

no formatting issue

**Quality:**

3

**Strengths And Weaknesses:**

Strengths:

This work is the first to use discrete flow matching to train unified models and it achieves promising results on multiple benchmarks.

The approach leverages metric-induced probability paths with kinetic optimal velocities for self-correction.

Weaknesses:

Discrete flow matching requires limiting the output length during inference compared with autoregressive methods, which makes it somewhat inconvenient for understanding tasks.

---

> ### Author Rebuttal · Authors · 2025-07-30
>
> Thanks for your great efforts on the review and constructive comments.
> We will try our best to answer all your questions.
>
> **Please let us know if you are not satisfied with our answers or have further questions, so that we can get back to you as soon as possible.**
>
> $\textcolor{blue}{\textsf{Q1: Ask about the fixed output length limitation of the discrete flow matching (DFM) framework.}}$
>
> > "Discrete flow matching requires limiting the output length during inference compared with autoregressive methods, which makes it somewhat inconvenient for understanding tasks."
>
> A1: Thank you for your careful review. We acknowledge that, in its standard form, the DFM framework requires specifying the output length during inference, in contrast to autoregressive models which naturally support variable-length outputs.
> However, recent advances—such as EditFlow [cite 1] and DreamOn [cite 2]—have introduced edit operations into the sequence generation process, enabling more flexible and dynamic adjustment of output lengths within non-autoregressive frameworks.
> We believe incorporating such approaches into FUDOKI is a promising direction for future work, which could further enhance its applicability to a wider range of understanding tasks.
>
>
> [cite 1] Havasi, Marton, et al. Edit Flows: Flow Matching with Edit Operations. Preprint, 2025.
>
> [cite 2] Wu, Zirui, et al. DreamOn: Diffusion Language Models For Code Infilling Beyond Fixed-size Canvas. Blog, 2025.
>
> $\textcolor{blue}{\textsf{Q2: Ask about the rationale for choosing Janus as the base model for FUDOKI.}}$
>
> > "Although the paper applies flow matching to unified models, it uses the AR-based Janus as the base model for training. Could the authors explain why they did not choose MaskGIT or another diffusion model instead?"
>
> A2: Thank you for your insightful question.
> We selected Janus as the base model because it demonstrates strong capabilities in both visual understanding and generation. In contrast, adopting MaskGIT or other diffusion-based generation models would require substantial retraining to achieve robust visual understanding, leading to significantly higher training costs. Furthermore, due to our computational constraints—all experiments were conducted on NVIDIA V100 GPUs in the paper—the compact size of Janus (under 2B parameters) makes it a more practical and efficient foundation for our framework.
>
>
> $\textcolor{blue}{\textsf{Q3: Ask about other applications of FUDOKI.}}$
>
> > "Beyond benchmark tests, it would strengthen the paper to include other applications. Could the authors provide results on interleaved image generation or on image editing tasks?"
>
> A3: Thank you for your question. We appreciate your interest in exploring broader applications of our framework.
> As an initial step, we conducted preliminary experiments on interleaved image-text generation, i.e., Maze Navigation, which involves one input image for understanding and one output image for generation. The results are presented in Figures 9–10 of the supplementary materials.
> However, due to computational constraints with V100 GPUs, we were unable to extend training to more complex multimodal scenarios involving multiple images at a large scale. We recognize the value of demonstrating FUDOKI's versatility and plan to explore advanced tasks such as image editing and large-scale interleaved generation in future work if more computational resources are available.

---

> > ### Comment · Reviewer_CdAa · 2025-08-04
> > **Reply to the rebuttal**
> >
> > I have read the rebuttal carefully, i will maintain my score.

---

### Comment · Area_Chair_seNE · 2025-08-04

Dear Reviewers,

This paper received a range of positive assessments prior to the rebuttal (i.e., Accept / Borderline Accept).

If you have any remaining concerns or points for clarification, please feel free to raise them and engage with the authors during the discussion period.

Thank you for your contributions!

---

### Note · Authors · 2025-08-16

We sincerely appreciate the reviewers’ time and effort in evaluating our paper and are pleased that they recognized several key strengths of our work:

### Novelty
- FUDOKI is the first multimodal model built on discrete flow matching (DFM), enabling bidirectional information flow and iterative self-correction beyond standard autoregressive (AR) decoding. [CdAa, Zj7H, sZnw, vmHC]
- Introduces metric-induced paths and kinetically optimal velocity fields for multimodal learning. [CdAa]
- Supports dynamic token revision during generation, improving flexibility. [sZnw, CdAa]

### Performance
- Strong results on visual understanding and generation benchmarks. [sZnw, Zj7H]
- Thorough evaluation across a wide range of established tasks and datasets. [Zj7H]

### Presentation
- Clear, well-structured, and solid presentation. [vmHC, Zj7H]
- Figures clearly highlight distinctions from existing approaches. [vmHC, Zj7H]

During the rebuttal phase, we conducted supplementary experiments:
- We evaluated more challenging mathematical reasoning on MathVista (testmini), where FUDOKI outperformed Janus-1.5B and Janus-Pro-1B at the same scale. [Zj7H]
- We assessed self-correction on MMVet using Janus-Pro-1B responses as baselines: FUDOKI was initialized with these baselines and refined them over 32 steps, achieving the largest gain, while AR-based models showed marginal or negative changes. [vmHC]


We thank the reviewers for their thoughtful and constructive feedback. We will incorporate the new experiments, additional discussions, citations, and clarifications into the revised manuscript.

---

### Decision · Program_Chairs · 2025-09-17

**Decision:**

Accept (spotlight)

**Comment:**

This paper introduces FUDOKI, the first unified multimodal model built entirely on discrete flow matching (DFM). Unlike autoregressive (AR) paradigms, FUDOKI enables bidirectional context modeling and iterative refinement, which is shown to improve reasoning and self-correction capabilities.

All reviewers leaned positively, praising the paper’s clarity, solid technical presentation, and strong empirical results. In particular, the rebuttal strengthened confidence by adding experiments on MathVista and MMVet, which demonstrated improved reasoning ability and self-correction—two aspects highly relevant to the field of multimodal large language models (MLLMs).

At the same time, reviewers noted certain limitations. The novelty is incremental in that FUDOKI extends prior DFM concepts rather than introducing an entirely new paradigm. The computational cost of training is significant, which may limit accessibility and deployment. Moreover, broader ablations and comparisons are missing, for instance against models such as Bagel, which would help position the contribution more clearly.

Overall, I agree with the reviewers that FUDOKI makes an original and timely contribution, showing convincingly that DFM can serve as a viable alternative to AR-based MLLMs. While there are concerns about efficiency, scope of evaluation, and related work coverage, these are issues the authors should address in a revision. On balance, the combination of a novel unified formulation, clear empirical strengths, and demonstrated reasoning improvements merits acceptance as a Spotlight.